# Development of ^18^F-Labeled Bispyridyl Tetrazines for In Vivo Pretargeted PET Imaging

**DOI:** 10.3390/ph15020245

**Published:** 2022-02-18

**Authors:** Rocío García-Vázquez, Jesper Tranekjær Jørgensen, Klas Erik Bratteby, Vladimir Shalgunov, Lars Hvass, Matthias M. Herth, Andreas Kjær, Umberto Maria Battisti

**Affiliations:** 1Department of Drug Design and Pharmacology, Faculty of Health and Medical Sciences, University of Copenhagen, Universitetsparken 2, 2100 Copenhagen, Denmark; rociogv@sund.ku.dk (R.G.-V.); klas.bratteby@sund.ku.dk (K.E.B.); vladimir.shalgunov@sund.ku.dk (V.S.); 2Department of Clinical Physiology, Nuclear Medicine & PET, Rigshospitalet, Blegdamsvej 9, 2100 Copenhagen, Denmark; 3Cluster for Molecular Imaging, Department of Biomedical Sciences, University of Copenhagen, Blegdamsvej 3, 2100 Copenhagen, Denmark; jespertj@sund.ku.dk (J.T.J.); lars.hvass@sund.ku.dk (L.H.); 4Department of Radiation Physics, Skåne University Hospital, Barngatan 3, 22242 Lund, Sweden

**Keywords:** bioorthogonal chemistry, tetrazine ligation, bispyridyl tetrazines, pretargeted imaging, PET, fluorine-18, molecular imaging

## Abstract

Pretargeted PET imaging is an emerging and fast-developing method to monitor immuno-oncology strategies. Currently, tetrazine ligation is considered the most promising bioorthogonal reaction for pretargeting in vivo. Recently, we have developed a method to ^18^F-label ultrareactive tetrazines by copper-mediated fluorinations. However, bispyridyl tetrazines—one of the most promising structures for in vivo pretargeted applications—were inaccessible using this strategy. We believed that our successful efforts to ^18^F-label H-tetrazines using low basic labeling conditions could also be used to label bispyridyl tetrazines via aliphatic nucleophilic substitution. Here, we report the first direct ^18^F-labeling of bispyridyl tetrazines, their optimization for in vivo use, as well as their successful application in pretargeted PET imaging. This strategy resulted in the design of [^18^F]**45**, which could be labeled in a satisfactorily radiochemical yield (RCY = 16%), molar activity (A_m_ = 57 GBq/µmol), and high radiochemical purity (RCP > 98%). The [^18^F]**45** displayed a target-to-background ratio comparable to previously successfully applied tracers for pretargeted imaging. This study showed that bispyridyl tetrazines can be developed into pretargeted imaging agents. These structures allow an easy chemical modification of ^18^F-labeled tetrazines, paving the road toward highly functionalized pretargeting tools. Moreover, bispyridyl tetrazines led to near-instant drug release of iTCO-tetrazine-based ‘click-to-release’ reactions. Consequently, ^18^F-labeled bispyridyl tetrazines bear the possibility to quantify such release in vivo in the future.

## 1. Introduction

Radioimmunoconjugates have emerged as important tools; e.g., in the diagnosis and treatment of cancer [1,2]. A subclass within these important substances are monoclonal antibodies (mAbs). They can be designed and produced with exquisite target affinity and selectivity. From a nuclear molecular imaging point of view, mAb-based agents can result in high target-to-background ratios with a low nondisplaceable binding component, which make them almost ideal tracers. Unfortunately, at the same time they possess slow pharmacokinetic properties; i.e., target accumulation and blood clearance takes days rather than hours [3,4]. Consequently, long-lived radionuclides must be used to match the pharmacokinetic profile of these vectors [3]. This results in unnecessary radiation burden for patients (Figure 1A), and sometimes even at a magnitude prohibitive for clinical studies [5]. An elegant way to overcome these limitations is a relatively new technique based on a pretargeted approach. In this approach, the accumulation timeframe of mAbs is temporally decoupled from the actual imaging process. In particular, a tagged mAb is administered days before a small molecule imaging agent is used to bioorthogonally react with the tag of the mAb. In this way, the accumulation of the mAb can be imaged even after days with the aid of a short-lived radionuclide (Figure 1B) [6].

Pretargeted imaging consequently exploits the unique targeting properties of mAbs and the rapid pharmacokinetics properties of small molecules, enabling exceptional target-to-background ratios within hours [3]. Currently, the most promising reaction for pretargeted imaging is the tetrazine ligation between a tetrazine (Tz) and a trans-cyclooctene (TCO) (Figure 1B). Several preclinical examples have been reported that demonstrated the potential of this reaction [6,7,8,9,10,11,12,13,14,15,16,17,18,19,20,21,22]. More importantly, in late 2020, the first clinical phase I trial based on tetrazine ligation was initiated (NCT04106492) [23]. From a clinical point of view, a fluorine-18 (^18^F)-labeled Tz would be ideal for positron emission tomography (PET) applications, since ^18^F possesses almost perfect physical characteristics for molecular imaging [24,25,26]. However, due to the intrinsic instability of the tetrazine scaffold to basic conditions—typically employed in direct ^18^F-fluorination approaches—no ^18^F-tetrazine was synthesized until few years ago [10,27,28,29,30,31]. Recently, we reported the first direct ^18^F-fluorination of highly reactive tetrazines, which could be labeled via copper-mediated oxidative ^18^F-fluorination [32]. This methodology allowed us to develop a low-lipophilicity, fast-clearing, and highly reactive tetrazine PET tracer (**1**) (Figure 2). In a similar way, we were able to develop the first direct aliphatic ^18^F-radiolabeled Tz (**2**) (Figure 2). This was possible using ultralow basic conditions [33,34]. Both compounds (**1** and **2**) showed high imaging contrast during in vivo experiments, and are currently subjects in further studies to evaluate their potential to be translated into the clinic.

However, both labeling methods (Cu-mediated ^18^F-fluorinartion and direct aliphatic ^18^F-labeling) have thus far only been successful for H-Tzs. Bispyridyl scaffolds could not be labeled with Cu-mediated approaches due to the chelating effect of the pyridyl nitrogen [32,35]. Aliphatic labeling was only applied on monosubstituted Tzs [34]. Bispyridyl Tzs are, however, very appealing subjects to be used for pretargeted experiments (Figure 2). Beside H-Tzs, these structures are the only other structural class that have been successfully applied in vivo [6,35,36,37,38]. Bispyridyl Tzs have intrinsic high-reaction kinetics and allow for greater flexibility with respect to structural modifications compared to H-Tzs. For example, one pyridyl unit can be used for labeling purposes, whereas the other can be used to introduce polarity or other functional scaffolds as NIR turn-on dyes or additional chelators for pretargeted radiotherapy [3,17,39,40]. Recently, bispyridyl Tzs also were shown to be superior for instant and near-quantitative ’click-to-release’ approaches [41]. Radiolabeled bispyridyl Tzs could be used as such to quantify drug release in vivo.

In this work, we describe the design, synthesis, and in vivo evaluation of the first ^18^F-labeled bispyridyl Tz. In order to develop such a Tz, we exploited one pyridyl ring to introduce polarity. We recently showed that a clogD_7_._4_ of at least -3 is a necessary parameter for successful in vivo pretargeting [17]. The other pyridyl ring was used for labeling purposes using a set of different linkers. The selection of the linker for the ^18^F-labeling was initially performed using H-Tzs due to their more accessible chemistry. Promising structures in term of radiochemical conversions (RCC) were translated into their bispyridyl Tz counterparts. Finally, the best candidate was applied for pretargeted PET experiments.

## 2. Results and Discussion

### 2.1. Evaluation of Different Fluoroethyl Linkers Using H-Tz as Model Structures

We have previously reported that fluoroethyl moieties can be used to introduce ^18^F-fluoride into H-Tz employing non-nucleophilic bases [33,34]. We believed that these conditions could also be suitable to label bispyridyl-based Tzs. In order to test the influence on the labeling step of different conjugation strategies between the fluoroethyl moiety and the Tz, we synthesized a set of structures possessing various linkers such as esters, amines, ethers and amides (Figure 3). We decided to use H-Tzs for this study, as they are easier to access. Seven different molecules (**3–9**) and their corresponding precursors were designed.

### 2.2. Synthesis of H-Tz Derivatives

Compound **8** and its precursor **8a** were synthesized as previously reported [33]. Compounds **3** and **4** and their corresponding precursor were obtained as reported in Figure 1.

Briefly, Tz **10** was synthesized from 4-cyanobenzoic acid using a Pinner-like, sulfur-mediated procedure in modest yields [42]. Subsequent alkylation with 1-fluoro-2-iodoethane or 2-bromoethanol respectively gave compounds **3** and **11**. The latter was then reacted with nosyl chloride to yield precursor **3a**. Coupling of 4-cyanobenzoic acid with 2-fluoroethylamine or ethanolamine afforded **12** and **13**; these were then converted into the corresponding Tz derivatives **4** and **14**. These intermediates were further nosylated under basic conditions. However, we were not able to isolate **4a,** since an intramolecular substitution occurred that could not be prevented. Tzs **5** and **6** and their respective precursors were obtained in a similar manner as reported in Figure 2. The yields obtained were comparable to those of **3** and **4**. However, the amide precursor **6a** could be isolated in a yield of 83%. This most likely was possible due the lower reactivity of the tetrazine core compared to that of **4a**.

Compounds **7** and **9** were obtained using a different synthesis strategy (Figure 3). 4-Hydroxybenzonitrile was reacted with 1-fluoro-2-iodoethane or 2-bromoethanol to respectively give **20** and **21** in 71% and 92% yields. These were converted into the corresponding Tzs **7** and **22** in modest yields (37% and 54%, respectively). The latter was further modified to give the nosylate precursor **7a**. Intermediates **17** and **18** were obtained from 4-(bromomethyl)benzonitrile and the corresponding amines. N-Boc protection and reaction with hydrazine gave Tzs **27** and **28**. Deprotection of **27** afforded compound **9** in an almost quantitative yield. Reaction of nosyl chloride with **28** resulted in an intramolecular reaction, and **9a** could not be obtained. For this reason, a different protective group was selected. Then, **28** was deprotected and reacted with trityl chloride to give **31**. In order to have the corresponding reference compound, **9** was converted to **30** as well. Finally, reaction with nosyl chloride gave **9b** in a yield of 56%.

### 2.3. Labeling of H-Tz Derivatives

As we recently reported, nosyl leaving groups are optimally suited to aliphatically ^18^F-label high-reactive Tzs. A mesylate or tosylate base precursor resulted in low radiochemical yields (RCY) [33,43]. Radiolabeling only succeeded using low basicity conditions, since H-Tzs are too sensitive for standard—rather, basic—aliphatic ^18^F-fluorination approaches [29,33]. Within this study, we applied the same labeling conditions. Furthermore, we also investigated to what extent the precursor concentration influenced the RCC. Two amounts (3.1 and 9.3 nmol) were selected in this respect. With the exception of compound **3**, both concentrations resulted in the same RCC range. Labeling attempts revealed further that the linkers influenced RCCs strongly. Only tetrazines **3**, **7,** and **8** could be synthesized. RCCs were in the range of 23–53% (based on radio-HPLC, *n* = 3). No or only minimal product formation could be observed with a different substitution profile. In light of that, bispyridyl Tzs **33**, **38,** and **41** were designed, synthesized, and then radiolabeled.

### 2.4. Synthesis of the Bispyridyl Tzs

Bispyridyl analogues of **3**, **7,** and **8** were synthesized as described in Figure 4. 6-Cyanonicotinic acid was converted to the corresponding bispyridyl Tz **32** following a previously reported procedure [44]. The latter was alkylated with 1-fluoro-2-iodoethane or 2-bromoethanol to respectively give compounds **33** and **34**. The alcohol derivative was then transformed to the nosylate precursor **33a**. Differently, 5-methylpicolinonitrile was brominated and reacted with ethylene glycol to afford **36**. Formation of the Tz ring and conversion to the fluorine analogue gave **38** [45]. Nosylation of the same alcohol intermediate yielded **38a**. The last compounds, **41** and **41a,** were obtained starting with 5-hydroxypicolinonitrile and using the same procedure employed for Tz **7**.

### 2.5. Labeling of Bispyridyl Tz Derivatives

Compounds [^18^F]**33**, [^18^F]**38,** and [^18^F]**41** were labeled from their corresponding precursors **33a**, **38a,** and **41a** following the same procedure reported in Section 2.3. The results are reported in Table 1. As expected from previous results, radiolabeling was successful in all cases. RCCs were in the range of 20–30%. However, only compounds [^18^F]**33** and [^18^F]**38** could be radiolabeled in a similar conversion when lower precursor concentrations were used. Due to that and its easier synthetic access compared to **33**, we decided to use **38** as a template for our next steps, aiming to develop a pretargeted imaging agent.

### 2.6. Synthesis and Evaluation of 45 by Ex Vivo Blocking Assay

Compound **45** was designed based on a fluorothexy moiety identified to yield in the highest RCCs of previous results in this study. To decrease lipophilicity, we further conjugated a diacetic acid moiety to the structure. This combination should have resulted in an easy-to-label compound, as well as an imaging agent with the necessary polarity to be applied for in vivo pretargeted imaging. The pathway to synthesize **45** is depicted in Figure 5. Alkylation of di-*tert*-butyl iminodiacetate by compound **35** under basic conditions yielded **43**. The latter was reacted with an excess of **39** and hydrazine hydrate to give **44**. Deprotection and purification by preparative-HPLC afforded **45** as a TFA salt.

Next, **45** was evaluated for its abilities in pretargeting. We recently developed a blocking assay that allowed us to assess the in vivo ligation performance of unlabeled tetrazine derivatives, omitting the time-consuming development of radiolabeled tetrazines for each tested ligand. It was based on the ability of Tzs to block the binding of the literature-known pretargeted imaging agent [^111^In]**47** to the pretargeting vector CC49-TCO (administered 72 h prior) in tumor-bearing mice [17,32,34]. In this setup, **45** showed almost complete blocking. This blocking effect was as good as the ones observed with currently successfully applied ‘state-of-the-art’ pretargeted imaging agents (Table 2). This result allowed us to assume that **45** could indeed be a suitable candidate for pretargeted imaging.

### 2.7. Synthesis and Labeling of [^18^F]***45***

The precursor (**45a**) was synthesized over 4 synthesis steps (Figure 5). Briefly, **40** was reacted with an excess of **43** and hydrazine hydrate to give, after oxidation, **46.** This was then converted to the corresponding nosylated derivative **45a**. Radiolabeling of [^18^F]**45** was carried out in a one-pot, two-step reaction sequence (Figure 4A,B). The conditions were similar to those we recently reported and applied for all former labeling attempts in this study [29,32,33,46]. A protection/deprotection strategy was needed to successfully label [^18^F]**45** in a radiochemical yield (RCY) of 16 ± 4% (*n* = 4), a radiochemical purity (RCP) of >98%, and a molar activity (A_M_) of 57 ± 15 GBq/µmol. The total synthesis was approximately 90 min, including labeling, purification, and formulation of the final product. The maximum isolated amount was 1.2–2 GBq (EOS) starting from ~17 GBq of fluoride-18. The [^18^F]**45** rapidly reacted with TCO-PNP carbonate, and was stable in PBS at room temperature for minimum of 4 h, as confirmed by radio-HPLC (Figure 4C).

### 2.8. In Vivo Pretargeting PET Imaging with [^18^F]45

The ability of [^18^F]**45** to act as a pretargeting imaging agent was evaluated in LS174T tumor-bearing mice. Mice were grouped and administered with either monoclonal anti-TAG72 antibody, CC49, or TCO-conjugated CC49 (CC49-TCO) 72 h prior to injection of [^18^F]**45**. PET/CT scans were acquired 1 h later, as well as image-derived uptake values from tumor, heart, and muscle tissue extracted by manually created regions of interests (Figure 5).

The tumors of the mice in the group receiving CC49-TCO had a significant higher uptake of [^18^F]**45** (1.8 ± 0.3%ID/mL, mean ± SEM, *n* = 5) compared to the animals in the control group (0.3 ± 0.1%ID/mL, *n* = 5). Uptake in muscle tissue was generally low, resulting in a tumor-to-muscle ratio of 8.3 in animals pretargeted with CC49-TCO. In contrast, the uptake in blood (heart used as a surrogate) in this group of animals was high, resulting in a tumor-to-blood (T/B) ratio of 0.8. The tumor-to-liver (T/L) ratio was 0.3. Therefore, the uptake and target-to-background contrast found for [^18^F]**45** were at comparable levels to previous findings [17,32,34]. However, there is a general need to improve the target-to-background contrast before clinical translation. This could either be achieved by developing primary and secondary vectors with improved performance, thereby increasing the tumor uptake. Further, the use of a clearing agent could also potentially accelerate clearance of the primary vector and thus improve the T/B ratio in the future [14].

## 3. Materials and Methods

### 3.1. Chemistry

#### 3.1.1. General

All reagents and solvents were purchased from commercial suppliers and used without further purification. Anhydrous tetrahydrofuran (THF) was obtained from an SG water solvent purification system (Pure Process Technology). Anhydrous dimethyl sulfoxide (DMSO), *N*,*N*-dimethylacetamide (DMA), MeCN, and pyridine were purchased from commercially suppliers and stored under argon. Reactions requiring anhydrous conditions were carried out under an inert atmosphere (nitrogen or argon) and using oven-dried glassware (152 °C). Syringes used to transfer anhydrous solvents or reagents were purged with argon prior to use. Other solvents were analytical or HPLC grade and were used as received. NMR spectra were acquired on a 600 MHz Bruker Avance III HD (600 MHz for ^1^H and 151 MHz for ^13^C), a 400 MHz Bruker Avance II (400 MHz for ^1^H, 376 MHz for ^19^F, and 101 MHz for ^13^C), and a 400 MHz Bruker Avance UltraShield (400 MHz for ^1^H, 376 MHz for ^19^F, and 101 MHz for ^13^C) using chloroform*-d*, MeOD, or DMSO*-d*_6_ as a deuterated solvent, and with the residual solvent as the internal reference. For all NMR experiments, the deuterated solvent signal was used as the internal lock. Coupling constants (*J* values) are given in Hertz (Hz). Multiplicities of ^1^H NMR signals are reported as follows: s, singlet; d, doublet; dd, doublet of doublets; ddd, doublet of doublets of doublets; dt, doublet of triplets; t, triplet; q, quartet; m, multiplet; br, broad signal. NMR spectra of all compounds were reprocessed in MestReNova software (version 12.0.22023) from the original FID files. Yields refer to isolated compounds estimated to be >90% pure as determined by ^1^H NMR (25 °C) and analytical HPLC (Please refer to the Appendix A). Analytical HPLC method: Thermo Fisher UltiMate 3000 with a C-18 column (Luna 5 μm C18(2) 100 Å, 150 mm × 4.6 mm). Eluents: A, H_2_O with 0.1% TFA; B, MeCN with 0.1% TFA. Gradient from 100% A to 100% B for 12 min, back to 100% A for 3 min, flow rate 2 mL/min. Detection by UV absorption at λ = 254 nm on a UVD 170U detector. Thin-layer chromatography (TLC) was carried out on silica gel 60 F_254_ plates from Merck (Germany). Visualization was accomplished by UV lamp (254 nm). Preparative HPLC was carried out on an UltiMate HPLC system (Thermo Scientific) consisting of an LPG-3200BX pump (10 mL/min), a Rheodyne 9725i injector, a 10 mL loop, a MWD-3000SD detector (254 nm), and an AFC-3000SD automated fraction collector, using a Gemini-NX C18 column (21.2 × 250 mm, 5 µm, 110Å) (Phenomenex) equipped with a guard. Purifications were performed using linear gradients of 0.1% TFA in MilliQ-H_2_O (A) and 0.1% TFA, 10% MilliQ-H_2_O in MeCN (B). Data were acquired and processed using Chromeleon software v. 6.80. Semipreparative HPLC was performed on the same system using a Luna 5µ C18 column (250 × 10 mm) with a flow rate of 3 mL/min. Automated flash column chromatography was performed on a CombiFlash NextGen 300+ system supplied by TeleDyne ISCO, equipped with RediSep silica-packed columns. Detection of the compounds was carried out by means of a UV–vis variable wavelength detector operating at 200 to 800 nm and by an evaporative light-scattering detector (ELSD). Solvent systems for separation were particular for each compound, but consisted of various mixtures of heptane, EtOAc, CH_2_Cl_2_, and MeOH. Microwave-assisted synthesis was carried out in a Biotage Initiator apparatus operating in single mode; the microwave cavity produced controlled irradiation at 2.45 GHz (Biotage AB, Uppsala, Sweden). The reactions were run in sealed vessels. These experiments were performed by employing magnetic stirring and a fixed hold time using variable power to reach (during 1–2 min) and then maintain the desired temperature in the vessel for the programmed time period. The temperature was monitored by an IR sensor focused on a point on the reactor vial glass. The IR sensor was calibrated to the internal solution’s reaction temperature by the manufacturer. Mass spectra analysis was completed using MS-Acquity-A: Waters Acquity UPLC with QDa-detector. CC49-TCO was kindly provided by Tagworks Pharmaceuticals, and it was obtained as previously described [14].

#### 3.1.2. 2-Fluoroethyl 4-(1,2,4,5-tetrazin-3-yl)benzoate (**3**)

4-(1,2,4,5-Tetrazin-3-yl)benzoic acid (**10**)

The compound was synthesized as previously reported [42]. 4-Cyanobenzoic acid (0.3 g, 2.00 mmol), CH_2_Cl_2_ (0.12 mL, 2.00 mmol), sulfur (0.12 g, 0.5 mmol), and ethanol (4.0 mL) were mixed together in a microwave reaction vial. Hydrazine monohydrate (0.82 mL, 16.00 mmol) was added dropwise with stirring. The vessel was sealed, and the reaction mixture was heated to 50 °C for 24 h. The reaction was diluted with 3 mL of CH_2_Cl_2_, and sodium nitrite (1.44 g, 20.00 mmol) in 20 mL of H_2_O was added dropwise to the mixture under cooling. Excess acetic acid (7 mL) was then added slowly, during which the solution turned bright red in color. The reaction mixture was extracted with CH_2_CL_2_ (3 × 20 mL). The organic phase was dried over MgSO_4_ and concentrated under reduced pressure. The resulting residue was purified using flash chromatography (CH_2_Cl_2_/MeOH 98/2) to give 0.08 g (20%) of the desired product as a pink solid. Rf = 0.31 (CH_2_Cl_2_/MeOH 95/5); ^1^H NMR (400 MHz, DMSO) δ 13.32 (s, 1H), 10.66 (s, 1H), 8.62 (d, *J* = 8.5 Hz, 2H), 8.22 (d, *J* = 8.5 Hz, 2H); ^13^C NMR (101 MHz, DMSO) δ 166.67, 165.08, 158.24, 135.70, 134.32, 130.20, 127.97.

2-Fluoroethyl 4-(1,2,4,5-tetrazin-3-yl)benzoate (**3**)

Compound **10** (0.04 g, 0.20 mmol) and 1-fluoro-2-iodoethane (0.05 g, 0.30 mmol) were dissolved in 2 mL dry DMF, and DIPEA (0.10 mL, 0.60 mmol) was added. The reaction was left at 70 °C overnight. The reaction was diluted with CH_2_Cl_2_ (15 mL) and washed with a saturated aqueous solution of NH_4_Cl (10 mL) and H_2_O (2 × 10 mL). The organic phase was dried over MgSO_4_ and concentrated under reduced pressure. Purification by flash chromatography (85/15 heptane/EtOAc) afforded 0.04 g (81%) of **3** as a red solid. Rf = 0.23 (heptane/EtOAc 80/20); ^1^H NMR (600 MHz, CDCl_3_) δ 10.21 (s, 1H), 8.65 (d, *J* = 8.6 Hz, 2H), 8.23 (d, *J* = 8.5 Hz, 2H), 4.89–4.72 (m, 1H), 4.69–4.64 (m, 1H), 4.63–4.57 (m, 1H), 4.57–4.50 (m, 1H); ^13^C NMR (101 MHz, CDCl_3_) δ 165.92, 165.50, 157.99, 135.75, 133.63, 130.58, 128.27, 81.26 (d, *J* = 171.1 Hz), 64.30 (d, *J* = 20.1 Hz).

#### 3.1.3. 2-(((4-Nitrophenyl)sulfonyl)oxy)ethyl 4-(1,2,4,5-tetrazin-3-yl)benzoate (**3a**)

2-Hydroxyethyl 4-(1,2,4,5-tetrazin-3-yl)benzoate (**11**)

Compound **10** (0.03 g, 0.15 mmol) and 2-bromoethanol (0.02 mL, 0.22 mmol) were dissolved in 3 mL dry DMF, DIPEA (0.08 mL, 0.44 mmol) was dissolved in 1 mL DMF and added dropwise, and the reaction was left at 70 °C. After the reaction was completed, it was cooled down, diluted with water (30 mL) and extracted with CH_2_Cl_2_ (3 × 20 mL). The crude was purified by flash chromatography (60/40 heptane/EtOAc) to give 0.03 g (82%) of the desired product as a red solid. Rf = 0.24 (heptane/EtOAc 70/30); ^1^H NMR (600 MHz, CDCl_3_) δ 10.28 (s, 1H), 8.72 (d, *J* = 8.1 Hz, 2H), 8.29 (d, *J* = 8.1 Hz, 2H), 4.54 (t, *J* = 4.6 Hz, 2H), 4.02 (t, *J* = 4.7 Hz, 2H), 2.56–1.97 (m, 1H), ^13^C NMR (151 MHz, CDCl_3_) δ 166.04, 165.93, 158.00, 135.68, 133.87, 130.55, 128.28, 67.10, 61.31.

2-(((4-Nitrophenyl)sulfonyl)oxy)ethyl 4-(1,2,4,5-tetrazin-3-yl)benzoate (**3a**)

To a solution of compound **11** (0.03 g, 0.12 mmol) and DIPEA (0.04 mL, 0.24 mmol) in CH_2_Cl_2_ (10 mL) were added nosyl chloride (0.041 g, 0.18 mmol) and DMAP (0.001 g, 0.01 mmol). The reaction was stirred at room temperature for 1 h. The solvent was removed under reduced pressure. Purification by flash chromatography (80/20 heptane/EtOAc) afforded 0.035 (67%) of **3a** as a red solid. Rf = 0.38 (heptane/EtOAc 70/30); ^1^H NMR (600 MHz, CDCl_3_) δ 10.31 (s, 1H), 8.71 (d, *J* = 8.5 Hz, 2H), 8.31 (d, *J* = 8.8 Hz, 2H), 8.16 (d, *J* = 8.5 Hz, 2H), 8.14–8.09 (m, 2H), 4.64–4.58 (m, 2H), 4.58–4.53 (m, 2H); ^13^C NMR (151 MHz, CDCl_3_) δ 165.77, 165.07, 158.06, 150.73, 141.65, 136.14, 132.92, 130.48, 129.17, 128.27, 124.50, 68.72, 62.28.

#### 3.1.4. N-(2-Fluoroethyl)-4-(1,2,4,5-tetrazin-3-yl)benzamide (**4**)

4-Cyano-N-(2-fluoroethyl)benzamide (**12**)

To a solution of 4-cyanobenzoic acid (0.73 g, 5.00 mmol) in CH_3_CN (20 mL) was added CDI (1.21 g, 7.50 mmol). The mixture was stirred at room temperature for 45 min, before addition of 2-fluoroethylamine hydrochloride (0.55 g, 5.50 mmol) and Et_3_N (2.09 mL, 15.00 mmol). The reaction mixture was stirred for 2 h. Water (30 mL) was added, and the mixture was extracted with EtOAc (3 × 20 mL). The organic phase was dried over MgSO_4_ and concentrated under reduced pressure to give 0.65 g (68%) of the desire compound as a white solid. Rf = 0.26 (heptane/EtOAc 60/40); ^1^H NMR (400 MHz, DMSO) δ 8.96 (s, 1H), 8.02 (d, *J* = 8.7 Hz, 2H), 7.97 (d, *J* = 8.6 Hz, 2H), 4.61 (t, *J* = 5.1 Hz, 1H), 4.50 (t, *J* = 5.1 Hz, 1H), 3.62 (q, *J* = 5.2 Hz, 1H), 3.56 (q, *J* = 5.2 Hz, 1H); ^13^C NMR (101 MHz, DMSO) δ 165.61, 138.57, 132.92, 128.55, 114.17, 82.49 (d, *J* = 165.8 Hz), 40.57 (d, *J* = 20.9 Hz).

N-(2-Fluoroethyl)-4-(1,2,4,5-tetrazin-3-yl)benzamide (**4**)

The compound was obtained from **12** (0.60 g, 3.12 mmol) as described for compound **10**. Purification by flash chromatography (60/40 heptane/EtOAc) afforded 0.36 g (47%) of **4** as a red solid. Rf = 0.38 (heptane/EtOAc 50/50); ^1^H NMR (400 MHz, DMSO) δ 10.65 (s, 1H), 8.95 (t, *J* = 5.5 Hz, 1H), 8.60 (d, *J* = 8.6 Hz, 2H), 8.15 (d, *J* = 8.6 Hz, 2H), 4.65 (t, *J* = 5.1 Hz, 1H), 4.53 (t, *J* = 5.1 Hz, 1H), 3.66 (q, *J* = 5.3 Hz, 1H), 3.59 (q, *J* = 5.2 Hz, 1H); ^13^C NMR (101 MHz, DMSO) δ 166.24, 165.59, 158.70, 138.21, 134.81, 128.75, 128.23, 82.57 (d, *J* = 165.8 Hz), 40.56 (d, *J* = 16.2 Hz).

#### 3.1.5. 2-(4-(1,2,4,5-Tetrazin-3-yl)benzamido)ethyl 4-nitrobenzenesulfonate (**4a**)

4-Cyano-N-(2-hydroxyethyl)benzamide (**13**)

The compound was obtained from 4-cyanobenzoic acid (1.17 g, 8.00 mmol) and ethanolamine (4.82 mL, 80.00 mmol) as described for compound **12**. Purification by flash chromatography (CH_2_Cl_2_/MeOH 95/5) afforded 1.37 g (90%) of the desired compound as a colorless oil. Rf = 0.22 (CH_2_Cl_2_/MeOH 95/5); ^1^H NMR (400 MHz, DMSO) δ 8.70 (t, *J* = 5.6 Hz, 1H), 8.01 (d, *J* = 8.6 Hz, 2H), 7.96 (d, *J* = 8.6 Hz, 2H), 4.75 (t, *J* = 5.6 Hz, 1H), 3.53 (q, *J* = 5.9 Hz, 2H), 3.39–3.32 (m, 2H); ^13^C NMR (101 MHz, DMSO) δ 165.38, 139.03, 132.83, 128.54, 118.83, 113.94, 60.01, 42.82.

N-(2-Hydroxyethyl)-4-(1,2,4,5-tetrazin-3-yl)benzamide (**14**)

The compound was obtained from **13** (1.00 g, 5.25 mmol) as described for compound **10**. Purification by flash chromatography (CH_2_Cl_2_/MeOH 97/3) afforded 0.39 g (30%) of the desired compound as a colorless oil. Rf = 0.31 (CH_2_Cl_2_/MeOH 95/5); ^1^H NMR (600 MHz, DMSO) δ 10.64 (s, 1H), 8.68 (t, *J* = 5.6 Hz, 1H), 8.61–8.56 (m, 2H), 8.16–8.10 (m, 2H), 4.75 (t, *J* = 5.6 Hz, 1H), 3.56 (q, *J* = 6.1 Hz, 2H), 3.38 (q, *J* = 6.0 Hz, 2H); ^13^C NMR (151 MHz, DMSO) δ 166.00, 165.61, 158.71, 158.68, 138.66, 128.72, 128.16, 60.13, 42.80.

2-(4-(1,2,4,5-Tetrazin-3-yl)phenyl)-4,5-dihydrooxazole

The compound was obtained from **14** (0.05 g, 0.21 mmol) as described for compound **3a**. An intramolecular substitution occurred during the reaction, giving this side product. Purification by flash chromatography (60/40 heptane/EtOAc) afforded 0.35 g (75%) of this side product as a red solid. Rf = 0.41 (heptane/EtOAc 50/50); ^1^H NMR (600 MHz, CDCl_3_) δ 10.18 (s, 1H), 9.09–8.44 (m, 2H), 8.11 (d, *J* = 8.1 Hz, 2H), 4.43 (t, *J* = 9.5 Hz, 2H), 4.06 (t, *J* = 9.5 Hz, 2H); ^13^C NMR (151 MHz, CDCl_3_) δ 166.07, 163.83, 157.88, 133.94, 132.05, 129.06, 128.22, 67.91, 55.17.

#### 3.1.6. 2-Fluoroethyl 2-(4-(1,2,4,5-tetrazin-3-yl)phenyl)acetate (**5**)

2-(4-(1,2,4,5-Tetrazin-3-yl)phenyl)acetic acid (**15**)

The compound was obtained from 2-(4-cyanophenyl)acetic acid (0.97 g, 6.00 mmol) as described for compound **10**. Purification by flash chromatography (98/2 CH_2_Cl_2_/MeOH) to afford 0.36 g (28%) of the desired compound as a pink solid. Rf = 0.19 (95/5 CH_2_Cl_2_/MeOH); ^1^H NMR (400 MHz, MeOD) δ 10.33 (s, 1H), 8.56 (d, *J* = 8.4 Hz, 2H), 7.59 (d, *J* = 8.3 Hz, 2H), 3.78 (s, 2H); ^13^C NMR (101 MHz, MeOD) δ 173.27, 166.24, 157.83, 140.15, 130.69, 130.13, 127.78, 40.39.

2-Fluoroethyl 2-(4-(1,2,4,5-tetrazin-3-yl)phenyl)acetate (**5**)

The compound was obtained from **15** (0.07 g, 0.32 mmol) as described for compound **3**. Purification by flash chromatography (80/20 heptane/EtOAc) afforded 0.042 g (49%) of **5** as a red solid. Rf = 0.20 (80/20 heptane/EtOAc); ^1^H NMR (400 MHz, CDCl_3_) δ 10.14 (s, 1H), 8.53 (d, *J* = 8.4 Hz, 2H), 7.47 (d, *J* = 8.1 Hz, 2H), 4.63–4.57 (m, 1H), 4.51–4.46 (m, 1H), 4.37–4.32 (m, 1H), 4.30–4.25 (m, 1H), 3.74 (s, 2H); ^13^C NMR (101 MHz, CDCl_3_) δ 170.58, 166.25, 157.79, 139.06, 130.62, 130.38, 128.56, 81.14 (d, *J* = 170.9 Hz), 63.96 (d, *J* = 20.2 Hz), 41.01.

#### 3.1.7. 2-(((4-Nitrophenyl)sulfonyl)oxy)ethyl 2-(4-(1,2,4,5-tetrazin-3-yl)phenyl)acetate (**5a**)

2-Hydroxyethyl 2-(4-(1,2,4,5-tetrazin-3-yl)phenyl)acetate (**16**)

The compound was obtained from **15** (0.07 g, 0.32 mmol) as described for compound **11**. Purification by flash chromatography (80/20 heptane/EtOAc) afforded 0.05 g (59%) of the desired product as a red solid. Rf = 0.17 (60/40 heptane/EtOAc); ^1^H NMR (400 MHz, CDCl_3_) δ 10.14 (s, 1H), 8.51 (d, *J* = 8.4 Hz, 2H), 7.46 (d, *J* = 8.3 Hz, 2H), 4.24–4.17 (m, 2H), 3.81–3.74 (m, 2H), 3.73 (s, 2H), 1.91 (s, 1H); ^13^C NMR (101 MHz, CDCl_3_) δ 171.09, 166.22, 157.78, 139.23, 130.58, 130.36, 128.55, 66.68, 61.06, 41.13.

2-(((4-Nitrophenyl)sulfonyl)oxy)ethyl 2-(4-(1,2,4,5-tetrazin-3-yl)phenyl)acetate (**5a**)

The compound was obtained from **16** (0.50 g, 0.19 mmol) as described for compound **3a**. Purification by flash chromatography (75/25 heptane/EtOAc) afforded 0.05 g (58%) of **5a** as a red solid. Rf = 0.27 (heptane/EtOAc 60/40); ^1^H NMR (400 MHz, CDCl_3_) δ 10.16 (s, 1H), 8.51 (d, *J* = 8.2 Hz, 2H), 8.30 (d, *J* = 8.8 Hz, 2H), 8.02 (d, *J* = 8.8 Hz, 2H), 7.43 (d, *J* = 8.2 Hz, 2H), 4.30 (s, 4H), 3.69 (s, 2H); ^13^C NMR (101 MHz, CDCl_3_) δ 170.30, 157.84, 150.86, 141.58, 138.66, 130.76, 130.35, 129.20, 128.56, 124.52, 68.35, 62.00, 40.86.

#### 3.1.8. 2-(4-(1,2,4,5-Tetrazin-3-yl)phenyl)-N-(2-fluoroethyl)acetamide (**6**)

2-(4-Cyanophenyl)-N-(2-fluoroethyl)acetamide (**17**)

The compound was obtained from 2-(4-cyanophenyl)acetic acid (0.5 g, 3.10 mmol) as described for compound **12** [17]. Concentration under reduced pressure afforded 0.46 g (72%) of the desired product as a white solid. Rf: 0.35 (heptane/EtOAc 30/70); ^1^H NMR (400 MHz, CDCl_3_) δ 7.57 (d, *J* = 8.3 Hz, 2H), 7.34 (d, *J* = 8.2 Hz, 2H), 5.88 (s, 1H), 4.47 (t, *J* = 4.8 Hz, 1H), 4.35 (t, *J* = 4.8 Hz, 1H), 3.59–3.49 (m, 3H), 3.46 (dt, *J* = 5.8, 4.7 Hz, 1H); ^13^C NMR (101 MHz, CDCl_3_) δ 169.58, 140.02, 132.56, 130.10, 118.58, 111.32, 82.48 (d, *J* = 166.7 Hz), 43.36, 40.23 (d, *J* = 19.5 Hz).

2-(4-(1,2,4,5-Tetrazin-3-yl)phenyl)-N-(2-fluoroethyl)acetamide (**6**)

The compound was obtained from **17** (0.40 g, 1.94 mmol) as described for compound **3 [17]**. Purification by flash chromatography (30/70 heptane/EtOAc) afforded 0.21 g (41%) of **6** as a red solid. Rf = 0.22 (heptane/EtOAc 40/60); ^1^H NMR (400 MHz, DMSO) δ 10.58 (s, 1H), 8.48–8.40 (m, 3H), 7.57 (d, *J* = 8.2 Hz, 2H), 4.51 (t, *J* = 5.0 Hz, 1H), 4.39 (t, *J* = 5.0 Hz, 1H), 3.62 (s, 2H), 3.43 (q, *J* = 5.3 Hz, 1H), 3.36 (q, *J* = 5.2 Hz, 1H); ^13^C NMR (101 MHz, DMSO) δ 170.27, 165.92, 158.54, 142.02, 130.61, 130.53, 128.15, 82.90 (d, *J* = 165.0 Hz), 42.56, 40.56 (d, *J* = 15.8 Hz).

#### 3.1.9. 2-(2-(4-(1,2,4,5-Tetrazin-3-yl)phenyl)acetamido)ethyl 4-nitrobenzenesulfonate (**6a**)

2-(4-Cyanophenyl)-N-(2-hydroxyethyl)acetamide (**18**)

The compound was obtained from 2-(4-cyanophenyl)acetic acid (2.1 g, 13.03 mmol) as described for compound **13**. Recrystallization from EtOAc afforded 1.60 g (60%) of the desired product as a white solid. Rf = 0.15 (heptane/EtOAc 20/80); ^1^H NMR (400 MHz, DMSO) δ 8.15 (s, 1H), 7.76 (d, *J* = 8.2 Hz, 2H), 7.46 (d, *J* = 8.0 Hz, 2H), 4.68 (t, *J* = 5.4 Hz, 1H), 3.54 (s, 2H), 3.40 (q, *J* = 5.8 Hz, 2H), 3.12 (q, *J* = 5.9 Hz, 2H); ^1^H NMR (400 MHz, DMSO) δ 8.15 (s, 1H), 7.76 (d, *J* = 8.2 Hz, 2H), 7.46 (d, *J* = 8.0 Hz, 2H), 4.68 (t, *J* = 5.4 Hz, 1H), 3.54 (s, 2H), 3.40 (q, *J* = 5.8 Hz, 2H), 3.12 (q, *J* = 5.9 Hz, 2H).

2-(4-(1,2,4,5-Tetrazin-3-yl)phenyl)-N-(2-hydroxyethyl)acetamide (**19**)

The compound was obtained from **18** (0.82 g, 4.00 mmol) as described for compound **3**. Purification by flash chromatography (98/2 CH_2_Cl_2_/MeOH) and recrystallization from EtOAc afforded 0.33 g (32%) of the desired product as a pink solid. Rf = 0.31 (CH_2_Cl_2_/MeOH 95/5); ^1^H NMR (400 MHz, DMSO) δ 10.31 (s, 1H), 8.54 (d, *J* = 8.2 Hz, 2H), 7.58 (d, *J* = 8.1 Hz, 2H), 3.67 (s, 2H), 3.62 (t, *J* = 5.8 Hz, 2H), 3.36–3.32 (m, 2H); ^13^C NMR (101 MHz, DMSO) δ 171.23, 165.43, 157.02, 140.25, 129.84, 129.01, 127.04, 59.30, 41.46, 40.99.

2-(2-(4-(1,2,4,5-Tetrazin-3-yl)phenyl)acetamido)ethyl 4-nitrobenzenesulfonate (**6a**)

The compound was obtained from **19** (0.06 g, 0.23 mmol) as described for compound **3a**. Purification by flash chromatography (99/1 CH_2_Cl_2_/MeOH) afforded 0.08 g (83%) of **3a** as a red solid. Rf = 0.31 (heptane/EtOAc 60/40); ^1^H NMR (400 MHz, DMSO) δ 10.51 (s, 1H), 8.46 (d, *J* = 8.9 Hz, 2H), 8.39 (d, *J* = 8.6 Hz, 2H), 8.29 (d, *J* = 8.9 Hz, 2H), 7.69 (d, *J* = 8.6 Hz, 2H), 5.83 (s, 1H), 4.29 (t, *J* = 7.0 Hz, 2H), 4.10 (t, *J* = 7.0 Hz, 2H), 3.31 (s, 2H); ^13^C NMR (101 MHz, DMSO) δ 165.77, 158.22, 151.32, 149.36, 141.57, 140.68, 129.72, 128.22, 128.11, 128.07, 125.43, 86.21, 67.35, 47.47.

#### 3.1.10. 3-(4-(2-Fluoroethoxy)phenyl)-1,2,4,5-tetrazine (**7**)

4-(2-Fluoroethoxy)benzonitrile (**20**)

To a solution of 4-hydroxybenzonitrile (0.6 g, 5.00 mmol) and K_2_CO_3_ (1.38 g, 10.00 mmol) in CH_3_CN (20 mL) was added 1-fluoro-2-iodoethane (1.04 g, 6.00 mmol). The reaction was refluxed for 12 h and then concentrated under reduced pressure. The resulting mixture was diluted with water (50 mL), extracted with EtOAc (3 × 50 mL), washed with brine (50 mL), dried over MgSO_4_, filtered, and concentrated under reduced pressure. Purification by flash chromatography (heptane/EtOAc 80/20) afforded 0.8 g (97%) of the desired compound as a yellow oil. Rf = 0.37 (heptane/EtOAc 70/30); ^1^H NMR (400 MHz, CDCl_3_) δ 7.62 (d, *J* = 8.9 Hz, 1H), 7.01 (d, *J* = 8.9 Hz, 1H), 5.05–4.79 (m, 1H), 4.77–4.71 (m, 1H), 4.36–4.29 (m, 1H), 4.29–4.22 (m, 1H); ^13^C NMR (101 MHz, CDCl_3_) δ 161.69, 134.06, 119.03, 115.33, 104.59, 81.48 (d, *J* = 171.7 Hz), 67.37 (d, *J* = 20.5 Hz).

3-(4-(2-Fluoroethoxy)phenyl)-1,2,4,5-tetrazine (**7**)

The compound was obtained from **20** (0.73 g, 4.42 mmol) as described for compound **3**. Purification by flash chromatography (heptane/EtOAc 90/10) afforded 0.36 g (37%) of the desired compound as a red oil. Rf = 0.33 (heptane/EtOAc 80/20); ^1^H NMR (400 MHz, CDCl_3_) δ 10.07 (s, 1H), 8.53 (d, *J* = 8.9 Hz, 2H), 7.06 (d, *J* = 8.9 Hz, 2H), 4.89–4.78 (m, 1H), 4.74–4.62 (m, 1H), 4.43–4.29 (m, 1H), 4.27–4.16 (m, 1H); ^13^C NMR (101 MHz, CDCl_3_) δ 166.06, 162.56, 157.40, 130.26, 124.56, 115.38, 81.63 (d, *J* = 171.5 Hz), 67.29 (d, *J* = 20.6 Hz).

#### 3.1.11. 2-(4-(1,2,4,5-Tetrazin-3-yl)phenoxy)ethyl 4-nitrobenzenesulfonate (**7a**)

4-(2-Hydroxyethoxy)benzonitrile (**21**)

To a solution of 4-hydroxybenzonitrile (1.42 g, 12.00 mmol) and K_2_CO_3_ (8.29 g, 60.00 mmol) in CH_3_CN (20 mL) was added 2-bromoethanol (2.55 mL, 36.00 mmol). The reaction was refluxed for 12 h and then concentrated under reduced pressure. The resulting mixture was diluted with water (50 mL), extracted with EtOAc (3 × 50 mL), washed with brine (50 mL), dried over MgSO_4_, filtered, and concentrated under reduced pressure. Purification by flash chromatography (heptane/EtOAc 40/60) afforded 1.41 g (72%) of the desired compound as a yellow oil. Rf = 0.18 (heptane/EtOAc 50/50); ^1^H NMR (400 MHz, DMSO) δ 7.76 (d, *J* = 8.8 Hz, 2H), 7.11 (d, *J* = 8.8 Hz, 2H), 4.92 (t, *J* = 5.5 Hz, 1H), 4.09 (t, *J* = 5.0 Hz, 2H), 3.73 (q, *J* = 5.0 Hz, 2H), ^13^C NMR (101 MHz, DMSO) δ 162.70, 134.63, 119.64, 116.05, 103.15, 70.56, 59.79.

2-(4-(1,2,4,5-Tetrazin-3-yl)phenoxy)ethan-1-ol (**22**)

The compound was obtained from **21** (0.98 g, 6.00 mmol) as described for compound **3**. Purification by flash chromatography (heptane/EtOAc 50/50) afforded 0.71 g (54%) of the desired product as a red solid. Rf = 0.27 (heptane/EtOAc 50/50); ^1^H NMR (400 MHz, DMSO) δ 10.50 (s, 1H), 8.47 (d, *J* = 9.0 Hz, 2H), 7.23 (d, *J* = 9.0 Hz, 2H), 4.94 (t, *J* = 5.5 Hz, 1H), 4.15 (t, *J* = 4.9 Hz, 2H), 3.78 (q, *J* = 5.2 Hz, 2H); ^13^C NMR (101 MHz, DMSO) δ 165.65, 163.10, 158.16, 130.10, 124.35, 115.95, 70.44, 59.92.

2-(4-(1,2,4,5-Tetrazin-3-yl)phenoxy)ethyl 4-nitrobenzenesulfonate (**7a**)

Compound **21** (0.10 g, 0.46 mmol) was mixed with nitrobenzenesulfonyl chloride (0.15 g, 0.69 mmol) and dissolved in 4 mL dry CH_2_Cl_2_ under argon. A mixture of DIPEA (0.33 mL, 1.83 mmol) and DMAP (5 mg, 0.04 mmol) in 1 mL dry CH_2_Cl_2_ was added at 0 °C under argon. The reaction was slowly heated to room temperature and left for 1 h. The reaction was diluted with 10 mL CH_2_Cl_2_ and washed with 20 mL NH_4_Cl (sat.) and H_2_O (2 × 20 mL). The organic phase was dried over N_2_SO_4_ and concentrated under reduced pressure. Purification by flash chromatography (heptane/EtOAc 60/40) afforded 0.12 g (65%) of the desired product as a red solid. Rf = 0.41 (heptane/EtOAc 50/50); ^1^H NMR (400 MHz, DMSO) δ 10.52 (s, 1H), 8.97–8.31 (m, 5H), 8.33–7.87 (m, 2H), 7.11 (d, *J* = 8.9 Hz, 2H), 4.97–4.50 (m, 2H), 4.48–4.06 (m, 2H); ^13^C NMR (101 MHz, DMSO) δ 165.57, 161.88, 158.28, 154.85, 147.71, 130.25, 127.38, 125.26, 123.79, 116.05, 75.01, 67.33.

#### 3.1.12. N-(4-(1,2,4,5-Tetrazin-3-yl)benzyl)-2-fluoroethan-1-amine (**9**)

4-(((2-Fluoroethyl)amino)methyl)benzonitrile (**23**)

To a solution of 4-(bromomethyl)benzonitrile (0.78 g, 4.00 mmol) in CH_3_CN (40 mL) was added K_2_CO_3_ (0.33 g, 24.0 mmol) and 2-fluoroethylamine hydrochloride (0.16 g, 16.0 mmol). The mixture was stirred at room temperature overnight. The solvent was removed under reduced pressure, and the residue was diluted with water (20 mL) and extracted with EtOAc. The combined organic layer was washed with brine, dried over MgSO_4_, filtered, and concentrated under reduced pressure. The crude product was purified by flash column chromatography using EtOAc (heptane/EtOAc 50/50) in heptane to afford 0.54 g (76%) of **23** as a colorless oil. Rf = 0.24 (heptane/EtOAc 40/60). ^1^H NMR (400 MHz, CDCl_3_) *δ* 7.55 (d, *J* = 8.2 Hz, 2H), 7.40 (d, *J* = 8.0 Hz, 2H), 4.63–4.48 (m, 1H), 4.47–4.37 (m, 1H), 3.84 (s, 2H), 2.93–2.84 (m, 1H), 2.84–2.72 (m, 1H), 1.65 (s, 1H). ^13^C NMR (101 MHz, CDCl_3_) *δ* 145.6, 132.3, 128.6, 118.9, 110.9, 83.5 (d, *J* = 165.5 Hz), 53.1, 49.1 (d, *J* = 19.7 Hz)

*Tert*-butyl 4-cyanobenzyl(2-fluoroethyl)carbamate (**25**)

To a solution of **23** (540 mg, 3.03 mmol) and Et_3_N (1.27 mL, 9.09 mmol) in CH_2_Cl_2_ (40 mL) was added Boc_2_O (790 mg, 3.63 mmol), and the mixture was stirred at room temperature for 12 h. The solution was washed with water and saturated K_2_CO_3_ solution, dried over Na_2_SO_4,_ filtered, and concentrated under reduced pressure. The crude product was purified by flash column chromatography using (heptane/EtOAc 70/30) to afford 0.710 g (84%) of the desired product as a colorless oil (mixture of rotamers). Rf = 0.42 (heptane/EtOAc 80/20). ^1^H NMR (400 MHz, CDCl_3_) *δ* 7.55 (d, *J* = 7.8 Hz, 2H), 7.27 (d, *J* = 7.8 Hz, 2H), 4.79–4.10 (m, 4H), 3.62–3.28 (m, 2H), 1.96–1.05 (m, 9H). ^13^C NMR (101 MHz, CDCl_3_) *δ* 155.4, 144.2, 143.8, 132.4, 128.1, 127.5, 118.7, 111.1, 83.2 (d, *J* = 168.2 Hz), 82.7 (d, *J* = 170.5 Hz), 52.1, 51.2, 47.7, 28.3.

*Tert*-butyl 4-(1,2,4,5-tetrazin-3-yl)benzyl(2-fluoroethyl)carbamate (**27**)

The compound was obtained from **25** (0.68 g, 2.44 mmol) as described for compound **3**. Purification by flash chromatography (heptane/EtOAc 80/20) afforded 0.18 g (22%) of the desired product as a red solid (mixture of rotamers). Rf = 0.21 (heptane/EtOAc 80/20); ^1^H NMR (400 MHz, CDCl_3_) δ 10.23 (s, 1H), 8.62 (d, *J* = 7.8 Hz, 2H), 7.49 (d, *J* = 7.8 Hz, 2H), 4.76–4.42 (m, 4H), 3.83–3.38 (m, 2H), 1.64–1.38 (m, 9H); ^13^C NMR (101 MHz, CDCl_3_) δ 166.32, 157.77, 155.51, 144.25, 132.37, 130.58, 128.55, 127.86, 83.46, 83.17 (d, *J* = 165.4 Hz), 82.67 (d, *J* = 170.4 Hz), 80.63, 52.07, 51.07, 47.49, 28.36.

N-(4-(1,2,4,5-tetrazin-3-yl)benzyl)-2-fluoroethan-1-amine (**9**)

To a solution of **27** (0.10 mg, 0.30 mmol) in dioxane (10 mL) was added a solution of HCl in dioxane (4.0 M, 3.0 mL). The mixture was stirred at room temperature for 2 h and then concentrated under reduced pressure. The obtained solid was washed with Et_2_O to afford 0.07 g (86%) of **9** as hydrochloride salt. ^1^H NMR (400 MHz, DMSO-*d*_6_) *δ* 10.64 (s, 1H), 9.87 (s, 2H), 8.55 (d, *J* = 8.3 Hz, 2H), 7.89 (d, *J* = 8.3 Hz, 2H), 4.89 (t, *J* = 4.6 Hz, 1H), 4.77 (t, *J* = 4.6 Hz, 1H), 4.35 (s, 2H), 3.38 (t, *J* = 4.7 Hz, 1H), 3.31 (t, *J* = 4.7 Hz, 1H). ^13^C NMR (101 MHz, DMSO-*d*_6_) *δ* 165.7, 158.7, 137.0, 132.8, 131.6, 128.4, 80.0 (d, *J* = 165.3 Hz), 50.1, 47.1 (d, *J* = 20.0 Hz).

#### 3.1.13. 2-((4-(1,2,4,5-Tetrazin-3-yl)benzyl)(*tert*-butoxycarbonyl)amino)ethyl 4-nitro Benzenesulfonate (**9a**)

4-(((2-Hydroxyethyl)amino)methyl)benzonitrile (**24**)

The compound was obtained from 4-(bromomethyl)benzonitrile (2.00 g, 10.20 mmol) and ethanolamine (12.30 mL, 200.00 mmol) as described for compound **23**. Concentration under reduced pressure afforded 1.65 g (92%) of the desired compound as a white solid. Rf = 0.15 (90/10 CH_2_Cl_2_/MeOH); ^1^H NMR (400 MHz, CDCl_3_) δ 7.62–7.52 (m, 2H), 7.46–7.37 (m, 2H), 3.88–3.77 (m, 2H), 3.69–3.59 (m, 2H), 2.79–2.70 (m, 2H), 2.58–2.23 (m, 1H); ^13^C NMR (101 MHz, CDCl_3_) δ 145.76, 132.22, 128.67, 118.88, 110.71, 60.91, 53.07, 50.78.

*Tert*-butyl (4-cyanobenzyl)(2-hydroxyethyl)carbamate (**26**)

The compound was obtained from **24** (1.62 g, 9.19 mmol) as described for compound **25**. Purification by flash chromatography (heptane/EtOAc 60/40) afforded 2.36 g (93%) of the desired product as a colorless oil (rotamers mixture). Rf = 0.4 (heptane/EtOAc 50/50); ^1^H NMR (600 MHz, CDCl_3_) δ 7.63 (d, *J* = 7.9 Hz, 2H), 4.55–4.52 (m, 2H), 3.74 (s, 2H), 3.51–2.94 (m, 2H), 2.69 (s, 1H), 1.76–0.53 (m, 9H); ^13^C NMR (151 MHz, CDCl_3_) δ 156.91, 144.11, 132.43, 128.02, 127.50, 118.72, 111.19, 81.02, 62.17, 61.46, 52.10, 51.17, 50.37, 49.51, 28.32.

*Tert*-butyl (4-(1,2,4,5-tetrazin-3-yl)benzyl)(2-hydroxyethyl)carbamate (**28**)

The compound was obtained from **26** (2.32 g, 8.32 mmol) as described for compound **3**. Purification by flash chromatography (80/20 heptane/EtOAc) to give 1.07 g (39%) of the desired product as a red oil (rotamers mixture). Rf. = 0.21 (heptane/EtOAc 70/30); ^1^H NMR (400 MHz, CDCl_3_) δ 10.17 (s, 1H), 8.53 (d, *J* = 8.0 Hz, 2H), 7.42 (d, *J* = 8.1 Hz, 2H), 4.57 (s, 2H), 3.71 (d, *J* = 5.8 Hz, 2H), 3.46–3.41 (m, 2H), 3.23 (q, *J* = 5.4 Hz, 1H), 1.39 (s, 9H); ^13^C NMR (101 MHz, CDCl_3_) δ 166.23, 157.75, 156.89, 144.17, 132.36, 130.55, 128.52, 127.89, 80.75, 62.36, 61.76, 51.98, 51.12, 50.13, 49.44, 28.34.

3-(4-(1,2,4,5-Tetrazin-3-yl)benzyl)oxazolidin-2-one

The compound was obtained from **28** (0.06 g, 0.18 mmol) as described for compound **3a**. An intramolecular substitution occurred during the reaction giving this side product. Purification by flash chromatography (50/50 heptane/EtOAc) afforded 0.035 (67%) of the side product as a red solid. Rf = 0.21 (heptane/EtOAc 50/50); ^1^H NMR (400 MHz, CDCl_3_) δ 10.16 (s, 1H), 8.55 (d, *J* = 8.3 Hz, 2H), 7.47 (d, *J* = 8.3 Hz, 2H), 4.46–3.83 (m, 2H), 3.45 (dd, *J* = 8.7, 7.3 Hz, 2H); ^13^C NMR (101 MHz, CDCl_3_) δ 166.16, 158.59, 157.88, 141.30, 131.38, 128.96, 128.83, 61.87, 48.23, 44.21.

#### 3.1.14. N-(4-(1,2,4,5-Tetrazin-3-yl)benzyl)-2-fluoro-N-tritylethan-1-amine (**30**)

Compound **9** (0.05 g, 0.18 mmol) was mixed with trityl chloride (0.05 g, 0.20 mmol) and dissolved in 4 mL dry CH_2_Cl_2_ under argon. A mixture of Et_3_N (0.08 mL, 0.55 mmol) in 1 mL dry CH_2_Cl_2_ was added at 0 °C under argon. The reaction was slowly heated to room temperature and left for 1 h. The reaction was diluted with 10 mL CH_2_Cl_2_ and washed with NH_4_Cl (sat.) 20 mL and H_2_O (2 × 20 mL). The organic phase was dried over N_2_SO_4_ and concentrated under reduced pressure. Purification by flash chromatography (heptane/EtOAc 80/20) afforded 0.08 g (78%) as a red solid. Rf = 0.41 (heptane/EtOAc 80/20); ^1^H NMR (600 MHz, CDCl_3_) δ 10.13 (s, 1H), 8.57 (d, *J* = 8.4 Hz, 2H), 7.77 (d, *J* = 8.4 Hz, 2H), 7.74–7.55 (m, 6H), 7.32–7.23 (m, 6H), 7.16–7.10 (m, 3H), 3.99 (t, *J* = 6.0 Hz, 1H), 3.91 (t, *J* = 6.1 Hz, 1H), 3.69 (s, 2H), 2.92–2.55 (m, 2H); ^13^C NMR (151 MHz, CDCl_3_) δ 166.46, 157.73, 147.36, 143.10, 130.32, 129.32, 128.56, 128.43, 127.93, 127.87, 126.46, 81.94 (d, *J* = 168.6 Hz), 79.00, 57.49, 54.66 (d, *J* = 22.8 Hz).

#### 3.1.15. 2-((4-(1,2,4,5-Tetrazin-3-yl)benzyl)(trityl)amino)ethyl 4-nitrobenzenesulfonate (**9b**)

2-((4-(1,2,4,5-Tetrazin-3-yl)benzyl)amino)ethan-1-ol (**29**)

To a solution of **28** (0.99 g, 2.98 mmol) in dioxane (10 mL) was added a solution of HCl in dioxane (4.0 M, 7.7 mL). The mixture was stirred at room temperature for 2 h and then concentrated under reduced pressure. The obtained solid was washed with Et_2_O to afford 0.72 g (90%) of **29** as hydrochloride salt. ^1^H NMR (400 MHz, DMSO) δ 10.64 (s, 1H), 9.50 (s, 3H), 8.96–8.31 (m, 2H), 8.09–7.60 (m, 2H), 4.32 (t, *J* = 5.7 Hz, 2H), 3.73 (t, *J* = 5.4 Hz, 2H), 3.01 (q, *J* = 5.6 Hz, 2H); ^13^C NMR (101 MHz, DMSO) δ 165.74, 158.72, 137.22, 132.77, 131.62, 128.37, 66.82, 56.83, 49.93, 49.16.

2-((4-(1,2,4,5-Tetrazin-3-yl)benzyl)(trityl)amino)ethan-1-ol (**31**)

The compound was obtained from **29** (0.05 g, 0.19 mmol) as described for compound **30**. Purification by flash chromatography (60/40 heptane/EtOAc) to give 0.04 g (48%) of the desired product as a red solid. Rf = 0.53 (60/40 heptane/EtOAc); ^1^H NMR (600 MHz, CDCl_3_) δ 10.22 (s, 1H), 8.65 (d, *J* = 8.2 Hz, 2H), 7.86 (d, *J* = 8.1 Hz, 2H), 7.69 (d, *J* = 7.9 Hz, 6H), 7.33 (t, *J* = 7.7 Hz, 6H), 7.22 (t, *J* = 7.3 Hz, 3H), 3.76 (s, 2H), 3.29 (t, *J* = 6.8 Hz, 2H), 2.56 (d, *J* = 7.0 Hz, 2H), 1.63 (s, 1H); ^13^C NMR (101 MHz, CDCl_3_) δ 166.40, 157.73, 147.86, 143.28, 130.35, 129.37, 128.55, 128.49, 126.38, 79.05, 61.76, 57.79, 57.12.

2-((4-(1,2,4,5-Tetrazin-3-yl)benzyl)(trityl)amino)ethyl 4-nitrobenzenesulfonate (**9b**)

The compound was obtained from **31** (0.04 g, 0.08 mmol) as described for compound **3a**. Purification by flash chromatography (90/10 heptane/EtOAc) to give 0.04 g (48%) of the desired product as a red solid. Rf = 0.25 (80/20 heptane/EtOAc); ^1^H NMR (400 MHz, CDCl_3_) δ 10.19 (s, 1H), 8.46 (d, *J* = 8.3 Hz, 2H), 8.08 (d, *J* = 8.8 Hz, 2H), 7.69 (d, *J* = 8.8 Hz, 2H), 7.64 (d, *J* = 8.2 Hz, 2H), 7.61–7.43 (m, 6H), 7.23 (t, *J* = 7.6 Hz, 6H), 7.16–7.09 (m, 3H), 4.32–3.39 (m, 4H), 2.57 (t, *J* = 7.2 Hz, 2H); ^13^C NMR (101 MHz, CDCl_3_) δ 166.20, 157.92, 150.45, 146.36, 142.60, 141.64, 130.79, 129.11, 128.98, 128.91, 128.50, 127.98, 126.69, 124.19, 79.05, 57.38.

#### 3.1.16. 2-Fluoroethyl 6-(4-(pyridin-2-yl)phenyl)nicotinate (**33**)

6-(6-(Pyridin-2-yl)-1,2,4,5-tetrazin-3-yl)nicotinic acid (**32**)

6-Cyanonicotinic acid (1.0 g, 6.75 mmol), 2-cyanopyridine (5.6 g, 45.01 mmol), and sulfur (0.43 g, 1.69 mmol) were suspended in EtOH (10 mL), followed by the addition of hydrazine hydrate (4.93 mL, 101.26 mmol). The reaction was heated to 90 °C for 2 h. The mixture was cooled to room temperature, and the formed precipitate was removed by filtration. Water (20 mL) and a solution of NaNO_2_ (9.31 g, 135.32 mmol) in 50 mL water were added, and the mixture was cautiously acidified to pH 2–3 by addition of AcOH. The mixture was extracted with CH_2_Cl_2_, and the combined organic layer was washed with water and brine, dried over MgSO_4_, and concentrated. The residue was purified by flash column chromatography (CH_2_Cl_2_/MeOH 95/5 + 0.1% AcOH) to afford 0.52 g (27%) of the desired compound as a pink solid. Rf = 0.21 (CH2Cl2/MeOH 95/5 + 0.1% AcOH); ^1^H NMR (400 MHz, DMSO) δ 9.37 (d, *J* = 2.1 Hz, 1H), 8.96 (q, *J* = 1.6 Hz, 1H), 8.71 (d, *J* = 8.1 Hz, 1H), 8.64 (d, *J* = 7.9 Hz, 1H), 8.58 (dd, *J* = 8.1, 2.1 Hz, 1H), 8.18 (td, *J* = 7.8, 1.8 Hz, 1H), 7.95–7.67 (m, 1H); ^13^C NMR (101 MHz, DMSO) δ 165.74, 163.21, 162.97, 153.13, 151.01, 150.70, 149.96, 138.66, 137.86, 128.69, 126.81, 124.53, 124.

2-Fluoroethyl 6-(6-(pyridin-2-yl)-1,2,4,5-tetrazin-3-yl)nicotinate (**33**)

The compound was obtained from 6-(6-(pyridin-2-yl)-1,2,4,5-tetrazin-3-yl)nicotinic acid (0.05 g, 0.18 mmol) as described for compound **3** to give 0.55 g (95%) of the desired product as a pink solid. Rf = 0.41 (heptane/EtOAc 20/80); ^1^H NMR (400 MHz, CDCl_3_) δ 9.56 (dd, *J* = 2.2, 0.9 Hz, 1H), 9.00 (ddd, *J* = 4.8, 1.8, 0.9 Hz, 1H), 8.85 (dd, *J* = 8.2, 0.9 Hz, 1H), 8.77 (dt, *J* = 7.9, 1.1 Hz, 1H), 8.02 (td, *J* = 7.8, 1.8 Hz, 1H), 7.60 (ddd, *J* = 7.7, 4.7, 1.2 Hz, 1H), 4.88–4.79 (m, 1H), 4.72 (td, *J* = 5.9, 3.6 Hz, 2H), 4.64 (dd, *J* = 5.0, 3.0 Hz, 1H); ^13^C NMR (101 MHz, CDCl_3_) δ 164.34, 163.84, 163.50, 151.97, 149.86, 137.55, 127.81, 124.78, 123.95, 81.04 (d, *J* = 171.5 Hz), 64.62 (d, *J* = 20.3 Hz).

#### 3.1.17. 2-(((4-Nitrophenyl)sulfonyl)oxy)ethyl 6-(6-(pyridin-2-yl)-1,2,4,5-tetrazin-3-yl)nicotinate (**33a**)

2-Hydroxyethyl 6-(6-(pyridin-2-yl)-1,2,4,5-tetrazin-3-yl)nicotinate (**34**)

The compound was obtained from **32** (0.04 g, 0.14 mmol) as described for compound **11**. The compound was used as a crude due to low solubility (0.035 g, 76%). Rf = 0.31 (CH_2_Cl_2_/MeOH 90/10); ^1^H NMR (400 MHz, CDCl_3_) δ 9.47 (dd, *J* = 2.1, 0.9 Hz, 1H), 9.01–8.90 (m, 1H), 8.86–8.74 (m, 1H), 8.70 (dd, *J* = 7.9, 1.1 Hz, 1H), 8.55 (dd, *J* = 8.2, 2.1 Hz, 1H), 7.96 (ddt, *J* = 10.4, 7.7, 2.2 Hz, 1H), 7.62–7.48 (m, 1H), 4.76–4.45 (m, 2H), 4.26–3.68 (m, 2H), 3.24 (s, 1H).

2-(((4-Nitrophenyl)sulfonyl)oxy)ethyl 6-(6-(pyridin-2-yl)-1,2,4,5-tetrazin-3-yl)nicotinate (**33a**)

The compound was obtained from **34** (0.035 g, 0.11 mmol) as reported for compound **3a**. Purification by flash chromatography (CH_2_Cl_2_/MeOH 95/5) afforded 0.035 g (64%) of the desired compound as a red solid. Rf = 0.35 (CH_2_Cl_2_/MeOH 95/5); ^1^H NMR (400 MHz, DMSO) δ 9.25 (d, *J* = 2.2 Hz, 1H), 8.76 (d, *J* = 8.4 Hz, 2H), 8.66 (d, *J* = 7.8 Hz, 1H), 8.50 (dd, *J* = 8.2, 2.2 Hz, 1H), 8.40 (d, *J* = 8.9 Hz, 2H), 8.24 (d, *J* = 8.9 Hz, 2H), 8.22–8.17 (m, 1H), 7.87–7.68 (m, 1H), 4.63 (s, 4H); ^13^C NMR (101 MHz, DMSO) δ 164.1, 163.71, 163.51, 154.88, 154.14, 151.23, 150.39, 141.06, 139.06, 138.39, 129.87, 127.37, 125.42, 125.08, 124.59, 123.78, 70.42, 63.28.

#### 3.1.18. 5-((2-Fluoroethoxy)methyl)-2-(4-(pyridin-2-yl)phenyl)pyridine (**38**)

5-(Bromomethyl)picolinonitrile (**35**)

To a solution of 5-methylpicolinonitrile (7.00 g, 59.25 mmol) and N-bromo succinimide (13.71 g, 77.03 mmol) in CH_3_CN (100 mL) was added AIBN (3.89 g, 23.70 mmol). The resulting solution was refluxed for 12 h. The reaction was cooled down, and EtOAc (200 mL) was added. The organic layer was washed with water (2 × 100 mL) and brine (2 × 100 mL), dried over anhydrous Na_2_SO_4_, filtered, and concentrated under reduced pressure. Purification by flash chromatography (n-heptane/EtOAc 90/10) afforded 7.1 g (61%) of **35** as a white solid. Rf = 0.29 (n-heptane/EtOAc 80/20); ^1^H NMR (400 MHz, CDCl_3_) δ 8.72 (d, *J* = 2.2 Hz, 1H), 7.87 (dd, *J* = 8.1, 2.3 Hz, 1H), 7.69 (dd, *J* = 8.1, 0.9 Hz, 1H), 4.49 (s, 2H); ^13^C NMR (101 MHz, CDCl_3_) δ 151.19, 137.50, 137.45, 133.40, 128.36, 116.84, 27.87.

5-((2-Hydroxyethoxy)methyl)picolinonitrile (**36**)

NaH (60% weight, 0.22 g, 5.58 mmol) was suspended in dry THF (10 mL), and ethylene glycol (2.83 mL, 50.75 mmol) was added dropwise under argon at 0 °C. The solution was left for 30 min at 0 °C before dropwise addition of 5-(bromomethyl)picolinonitrile (1.00 g, 5.87 mmol) in dry THF (10 mL) under argon at 0 °C. The reaction was stirred to room temperature for 10 min and refluxed for 3 h. The reaction was cooled to room temperature and quenched by adding EtOAc (40 mL), and the crude was washed with NH_4_Cl (sat, 50 mL × 2) and water (50 mL). The organic phase was dried over Na_2_SO_4_ and concentrated under reduced pressure. Purification by flash chromatography (DCM/MeOH 98/2) afforded 0.55 g (61%) of the desired product as a yellow oil. Rf = 0.31 (heptane/EtOAc 30/70); ^1^H NMR (400 MHz, CDCl_3_) δ 8.60 (d, *J* = 2.2 Hz, 1H), 7.82 (dd, *J* = 8.1, 2.2 Hz, 1H), 7.95–7.32 (m, 1H), 4.61 (s, 2H), 3.74 (dd, *J* = 5.4, 3.8 Hz, 2H), 3.61 (dd, *J* = 5.4, 3.8 Hz, 2H); ^13^C NMR (101 MHz, CDCl_3_) δ 149.94, 138.24, 135.89, 132.41, 128.30, 117.20, 72.47, 69.77, 61.51.

2-((6-(6-(Pyridin-2-yl)-1,2,4,5-tetrazin-3-yl)pyridin-3-yl)methoxy)ethan-1-ol (**37**)

Compound **36** (0.55 g, 3.08 mmol), 2-cyanopyridine (1.6 g, 15.43 mmol), and sulfur (0.2 g, 0.77 mmol) were suspended in EtOH (5 mL), followed by the addition of hydrazine hydrate (2.26 mL, 43.3 mmol). The reaction was heated to 90 °C for 2 h. The mixture was cooled to room temperature, and the formed precipitate was removed by filtration. Water (20 mL) and a solution of NaNO_2_ (4.26 g, 61.73 mmol) in 30 mL water were added, and the mixture was cautiously acidified to pH 2 by addition of AcOH. The mixture was extracted with CH_2_Cl_2_, and the combined organic layer was washed with water and brine, dried over MgSO_4_, and concentrated. The residue was purified by flash column chromatography (CH_2_Cl_2_/MeOH 95/5) to afford 0.4 g (42%) of the desired compound as a pink solid. Rf = 0.21 (CH_2_Cl_2_/MeOH 95/5); ^1^H NMR (400 MHz, CDCl_3_) δ 9.03–8.94 (m, 1H), 8.90 (d, *J* = 2.1 Hz, 1H), 8.79–8.65 (m, 2H), 8.00 (ddd, *J* = 7.9, 5.5, 2.1 Hz, 3H), 7.57 (ddd, *J* = 7.6, 4.7, 1.2 Hz, 1H), 4.74 (s, 2H), 3.88–3.78 (m, 2H), 3.70 (dd, *J* = 5.3, 3.9 Hz, 2H), 2.76 (s, 1H); ^13^C NMR (101 MHz, CDCl_3_) δ 163.79, 163.69, 150.99, 150.03, 149.30, 137.50, 137.21, 136.47, 126.59, 124.50, 124.25, 72.24, 70.26, 61.80.

3-(5-((2-Fluoroethoxy)methyl)pyridin-2-yl)-6-(pyridin-2-yl)-1,2,4,5-tetrazine (**38**)

The compound was obtained from 2-((6-(6-(pyridin-2-yl)-1,2,4,5-tetrazin-3-yl)pyridin-3-yl)methoxy)ethan-1-ol (0.1 g, 0.33 mmol) dissolved in 5 mL dry THF and PBSF (0.20 mg, 0.66 mmol), and DIPEA (0.34 mL, 1.93 mmol) was added. Et_3_N.3HF (0.11 mL, 0.66 mmol) was dissolved in 2 mL THF and added dropwise. The reaction was left at room temperature for 12 h. The reaction was diluted with 20 mL CH_2_Cl_2_ and washed with NH_4_Cl (sat). The aqueous phase was extracted with CH_2_Cl_2_ (2 × 10 mL), and the combined organic layers were dried over MgSO_4_ and concentrated under reduced pressure. Purification by flash chromatography (CH_2_Cl_2_/MeOH 95/5) afforded 0.075 g (75%) of the desired product as a pink solid. Rf = 0.45 (CH_2_Cl_2_/MeOH 95/5); ^1^H NMR (400 MHz, CDCl_3_) δ 9.05–8.80 (m, 2H), 8.78–8.58 (m, 2H), 8.21–7.86 (m, 2H), 7.51 (ddd, *J* = 7.6, 4.7, 1.2 Hz, 1H), 4.71 (s, 2H), 4.65–4.58 (m, 1H), 4.57–4.42 (m, 1H), 3.97–3.78 (m, 1H), 3.77–3.68 (m, 1H); ^13^C NMR (101 MHz, CDCl_3_) δ 163.82, 163.73, 150.98, 150.03, 149.97, 149.36, 137.50, 137.03, 136.43, 126.60, 124.48, 124.26, 82.98 (d, *J* = 169.5 Hz), 70.35, 70.00 (d, *J* = 19.6 Hz).

#### 3.1.19. 2-((6-(4-(Pyridin-2-yl)phenyl)pyridin-3-yl)methoxy)ethyl 4-methylbenzene sulfonate (**38a**)

The compound was obtained from **37** ((0.05 g, 0.17 mmol) as reported for compound **3a**. Purification by flash chromatography (CH_2_Cl_2_/MeOH 98/2) afforded 0.07 g (84%) of the desired compound as a red oil. Rf = 0.48 (CH_2_Cl_2_/MeOH 90/10); ^1^H NMR (600 MHz, CDCl_3_) δ 9.02 (dt, *J* = 4.5, 1.4 Hz, 1H), 8.86 (d, *J* = 2.1 Hz, 1H), 8.80–8.63 (m, 2H), 8.41–8.33 (m, 1H), 8.17–8.11 (m, 1H), 8.04 (td, *J* = 7.7, 1.8 Hz, 1H), 7.93 (dd, *J* = 8.0, 2.2 Hz, 1H), 7.61 (ddd, *J* = 7.6, 4.7, 1.1 Hz, 1H), 4.69 (s, 2H), 4.44–4.38 (m, 2H), 3.90–3.82 (m, 2H); ^13^C NMR (151 MHz, CDCl_3_) δ 163.90, 163.68, 151.09, 150.76, 150.11, 150.00, 149.76, 141.81, 137.50, 136.40, 136.21, 129.25, 126.62, 124.58, 124.41, 124.22, 70.44, 70.15, 68.20.

#### 3.1.20. 5-(2-Fluoroethoxy)-2-(4-(pyridin-2-yl)phenyl)pyridine (**41**)

5-(2-Fluoroethoxy)picolinonitrile (**39**)

The compound was obtained from 5-hydroxypicolinonitrile (0.5 g, 4.16 mmol) as described for compound **20** to give 0.56 g (81%) of the desired product as a pink solid. Rf = 0.34 (heptane/EtOAc 70/30); ^1^H NMR (400 MHz, CDCl_3_) δ 8.34 (d, *J* = 2.9 Hz, 1H), 7.59 (d, *J* = 8.6 Hz, 1H), 7.27–7.20 (m, 1H), 4.83–4.76 (m, 1H), 4.71–4.64 (m, 1H), 4.35–4.28 (m, 1H), 4.28–4.21 (m, 1H); ^13^C NMR (101 MHz, CDCl_3_) δ 155.04, 138.50, 127.74, 123.96, 118.83, 115.55, 79.47 (d, *J* = 172.4 Hz), 66.05 (d, *J* = 20.4 Hz).

3-(5-(2-Fluoroethoxy)pyridin-2-yl)-6-(pyridin-2-yl)-1,2,4,5-tetrazine (**41**)

The compound was obtained from **39** (0.55 g, 3.31 mmol) as described for compound **37,** followed by purification by preparative HPLC to give 0.04 g (4%) of the desired product as a pink solid. Rf = 0.32 (CH_2_Cl_2_/MeOH 97/3); ^1^H NMR (400 MHz, CDCl_3_) δ 8.94 (dt, *J* = 4.7, 1.4 Hz, 1H), 8.70 (d, *J* = 8.5 Hz, 2H), 8.65 (d, *J* = 2.9 Hz, 1H), 8.00 (td, *J* = 7.8, 1.7 Hz, 1H), 7.57 (ddd, *J* = 7.7, 4.8, 1.2 Hz, 1H), 7.46 (dd, *J* = 8.8, 2.9 Hz, 1H), 4.88–4.81 (m, 1H), 4.76–4.69 (m, 1H), 4.44–4.38 (m, 1H), 4.38–4.25 (m, 1H); ^13^C NMR (101 MHz, CDCl_3_) δ 163.26, 163.21, 157.38, 150.46, 149.69, 142.33, 139.43, 138.19, 126.84, 125.92 124.58, 121.99, 81.39 (d, *J* = 172.7 Hz), 67.99 (d, *J* = 20.4 Hz); HPLC-MS [M+H]+ *m/z* calc. for [C_14_H_12_FN_6_O]+: 299.10; found: 299.12.

#### 3.1.21. 2-((6-(4-(Pyridin-2-yl)phenyl)pyridin-3-yl)oxy)ethyl 4-nitrobenzenesulfonate (**41a**)

5-(2-Hydroxyethoxy)picolinonitrile (**40**)

The compound was obtained from 5-hydroxypicolinonitrile (1.4 g, 11.65 mmol) as described for compound **21**. Purification by flash chromatography (n-heptane/EtOAc 40/60) afforded 1.45 g (76%) of the desired product as a white solid. Rf = 0.21 (heptane/EtOAc 40/60); ^1^H NMR (400 MHz, CDCl_3_) δ 8.32 (d, J = 2.9 Hz, 1H), 7.59 (d, J = 8.6 Hz, 1H), 7.42–6.71 (m, 1H), 4.66–4.05 (m, 2H), 3.97 (dd, J = 5.1, 3.9 Hz, 2H), 2.27 (s, 1H); ^13^C NMR (101 MHz, CDCl_3_) δ 157.27, 140.42, 129.65, 125.41, 120.58, 117.39, 70.21, 60.86.

2-((6-(6-(Pyridin-2-yl)-1,2,4,5-tetrazin-3-yl)pyridin-3-yl)oxy)ethan-1-ol (**42**)

The compound was obtained from **40** (0.5 g, 3.04 mmol) as described for compound **37**. Purification by flash chromatography (CH_2_Cl_2_/MeOH 97/3) afforded 0.27 g (30%) of the desired product as a pink solid. Rf = 0.25 (CH_2_Cl_2_/MeOH 95/5); ^1^H NMR (600 MHz, DMSO) δ 8.93 (ddd, *J* = 4.7, 1.8, 0.9 Hz, 1H), 8.65 (dd, *J* = 2.9, 0.6 Hz, 1H), 8.62–8.49 (m, 2H), 8.16 (td, *J* = 7.7, 1.8 Hz, 1H), 7.82–7.56 (m, 2H), 5.00 (s, 1H), 4.50–3.97 (m, 2H), 3.81 (td, *J* = 5.4, 4.3 Hz, 2H); ^13^C NMR (151 MHz, DMSO) δ 163.49, 163.30, 157.81, 151.05, 150.71, 142.37, 139.97, 138.25, 127.00, 126.05, 124.59, 121.95, 70.96, 59.88.

2-((6-(4-(Pyridin-2-yl)phenyl)pyridin-3-yl)oxy)ethyl 4-nitrobenzenesulfonate (**41a**)

The compound was obtained from **42** (0.05 g, 0.17 mmol) as reported for compound **3a**. Purification by flash chromatography (CH_2_Cl_2_/MeOH 98/2) afforded 0.03 g (37%) of the desired compound as a red solid (mixture of conformers 70/30). Rf = 0.61 (CH_2_Cl_2_/MeOH 90/10); ^1^H NMR (600 MHz, DMSO) δ 8.94 (ddd, *J* = 4.7, 1.8, 0.9 Hz, 1H), 8.70 (d, *J* = 2.9 Hz, 0.3H), 8.65 (d, *J* = 8.5 Hz, 0.3H), 8.60 (d, *J* = 7.8 Hz, 1H), 8.56 (d, *J* = 8.8 Hz, 0.7H), 8.51 (d, *J* = 2.9 Hz, 0.7H), 8.47–8.43 (m, 1.4H), 8.27–8.22 (m, 1.4H), 8.21–8.18 (m, 0.6H), 8.16 (td, *J* = 7.8, 1.8 Hz, 1H), 7.90–7.81 (m, 0.6H), 7.78 (dd, *J* = 8.8, 2.9 Hz, 0.3H), 7.73 (ddd, *J* = 7.6, 4.7, 1.2 Hz, 1H), 7.64 (dd, *J* = 8.8, 3.0 Hz, 0.7H), 4.81–4.71 (m, 0.6H), 4.68–4.59 (m, 1.4H), 4.57–4.54 (m, 0.6H), 4.51–4.43 (m, 1.4H); ^13^C NMR (151 MHz, DMSO) δ 163.57, 163.53, 163.26, 163.21, 156.74, 156.60, 154.89, 151.16, 151.08, 150.66, 147.70, 143.22, 142.95, 140.95, 139.86, 139.77, 138.28, 129.93, 127.38, 127.08, 127.05, 126.08, 125.89, 125.43, 124.63, 123.78, 122.36, 122.12, 74.80, 70.58, 67.80, 66.45.

#### 3.1.22. 2,2′-(((6-(6-(5-(2-Fluoroethoxy)pyridin-2-yl)-1,2,4,5-tetrazin-3-yl)pyridin-3-yl)methyl)azanediyl)diacetic Acid (**45**)

Di-*tert*-butyl 2,2′-(((6-cyanopyridin-3-yl)methyl)azanediyl)diacetate (**43**)

To a solution of 5-(bromomethyl)picolinonitrile (1.5 g, 7.61 mmol) in CH_3_CN (50 mL) was added K_2_CO_3_ (3.15 g, 22.84 mmol) and di-*tert*-butyl iminodiacetate (1.96 g, 7.99 mmol). The reaction mixture was stirred at room temperature overnight, and then the solvent was concentrated under reduced pressure. The resulting mixture was diluted with water (100 mL), extracted with EtOAc (3 × 40 mL), washed with brine (50 mL), dried over MgSO_4_, filtered, and concentrated under reduced pressure. Purification by flash chromatography (n-heptane/EtOAc 90/10) afforded 2.7 g (98%) of the desired compound as a white solid. Rf = 0.33 (n-heptane/EtOAc 80/20); ^1^H NMR (400 MHz, CDCl_3_) δ 8.61 (dd, *J* = 2.1, 0.9 Hz, 1H), 7.96 (dd, *J* = 8.0, 2.1 Hz, 1H), 7.60 (dd, *J* = 7.9, 0.9 Hz, 1H), 3.93 (s, 2H), 3.33 (s, 4H), 1.40 (s, 18H); ^13^C NMR (101 MHz, CDCl_3_) δ 170.09, 151.41, 138.96, 137.51, 132.77, 128.27, 117.35, 81.52, 55.41, 54.57, 28.16.

Di-*tert*-butyl 2,2′-(((6-(6-(5-(2-fluoroethoxy)pyridin-2-yl)-1,2,4,5-tetrazin-3-yl)pyridin-3-yl) methyl)-azanediyl)diacetate (**44**)

Compound **43** (0.2 g, 0.55 mmol), compound **39** (0.46 g, 2.77 mmol), and sulfur (0.36 g, 0.14 mmol) were suspended in EtOH (5 mL), followed by the addition of hydrazine hydrate (0.40 mL, 8.31 mmol). The reaction was heated to 90 °C for 2 h. The mixture was cooled to room temperature and the formed precipitate was removed by filtration. Water (20 mL) and a solution of NaNO_2_ (0.76 g, 11.07 mmol) in 10 mL water were added, and the mixture was cautiously acidified to pH 2–3 by addition of AcOH. The mixture was extracted with CH_2_Cl_2_, and the combined organic layer was washed with water and brine, dried over MgSO_4_, and concentrated. The residue was purified by flash column chromatography (CH_2_Cl_2_/MeOH 95/5) and crystallized from MeOH to afford 0.07 g (23%) of the desired compound as a pink solid. Rf = 0.36 (CH_2_Cl_2_/MeOH 95/5); ^1^H NMR (600 MHz, DMSO) δ 8.86 (d, *J* = 2.1 Hz, 1H), 8.68 (dd, *J* = 7.9, 3.1 Hz, 1H), 8.62 (d, *J* = 8.8 Hz, 1H), 8.58 (d, *J* = 8.0 Hz, 1H), 8.11 (dd, *J* = 8.1, 2.1 Hz, 1H), 7.76 (dd, *J* = 8.8, 3.0 Hz, 1H), 4.97–4.86 (m, 1H), 4.81 (dd, *J* = 4.8, 2.8 Hz, 1H), 4.54 (dd, *J* = 4.8, 2.7 Hz, 1H), 4.51–4.43 (m, 1H), 4.01 (s, 2H), 3.45 (s, 4H), 1.43 (s, 18H); ^13^C NMR (151 MHz, DMSO) δ 170.35, 163.47, 163.24, 157.21, 151.20, 149.63, 142.89, 139.86, 138.19, 138.11, 126.00, 124.30, 82.43 (d, *J* = 166.8 Hz), 80.97, 68.39 (d, *J* = 18.9 Hz), 55.57, 54.87, 28.29.

2,2′-(((6-(6-(5-(2-Fluoroethoxy)pyridin-2-yl)-1,2,4,5-tetrazin-3-yl)pyridin-3-yl)methyl)azane-diyl)diacetic acid (**45**)

To a solution of **44** (0.006 g, 0.83 mmol) in 5 mL of CH_2_Cl_2_ was added 2 mL of TFA. The mixture was stirred at room temperature for 4 h. The solvent was then removed under reduced pressure. Purification by preparative HPLC afforded 0.35 g (58%) of the desired compound (TFA salt) as a red solid. ^1^H NMR (600 MHz, DMSO) δ 8.89 (d, *J* = 2.1 Hz, 1H), 8.68 (d, *J* = 2.9 Hz, 1H), 8.62 (d, *J* = 8.8 Hz, 1H), 8.57 (dd, *J* = 8.0, 0.8 Hz, 1H), 8.14 (dd, *J* = 8.1, 2.1 Hz, 1H), 7.76 (dd, *J* = 8.8, 3.0 Hz, 1H), 4.91–4.86 (m, 1H), 4.83–4.79 (m, 1H), 4.57–4.52 (m, 1H), 4.52–4.47 (m, 1H), 4.06 (s, 2H), 3.53 (s, 4H); ^13^C NMR (151 MHz, DMSO) δ 172.62, 163.47, 163.23, 157.22, 151.33, 149.58, 142.89, 139.86, 138.28, 126.00, 124.27, 122.16, 82.43 (d, *J* = 166.8 Hz), 68.39 (d, *J* = 18.8 Hz), 55.00, 54.49; HPLC-MS [M+H]+ *m/z* calc. for [C_19_H_19_FN_7_O_5_]+: 444.14; found: 444.13.

#### 3.1.23. Di-*tert*-butyl 2,2′-(((6-(6-(5-(2-(((4-nitrophenyl)sulfonyl)oxy)ethoxy)pyridin-2-yl)-1,2,4,5-tetrazin-3-yl)pyridin-3-yl)methyl)azanediyl)diacetate (**45a**)

Di-*tert*-butyl 2,2′-(((6-(6-(5-(2-hydroxyethoxy)pyridin-2-yl)-1,2,4,5-tetrazin-3-yl)pyridin-3-yl)methyl)azanediyl)diacetate (**46**)

Compound **43** (1.05 g, 2.92 mmol), compound **40** (0.12 g, 0.73 mmol), and sulfur (0.05 g, 0.18 mmol) were suspended in EtOH (3 mL), followed by the addition of hydrazine hydrate (0.53 mL, 10.96 mmol). The reaction was heated to 90 °C for 2 h. The mixture was cooled to room temperature and the formed precipitate was removed by filtration. Water (10 mL) and a solution of NaNO_2_ (1.0 g, 14.62 mmol) in 10 mL water were added, and the mixture was cautiously acidified to pH 2 by addition of AcOH. The mixture was extracted with CH_2_Cl_2_, and the combined organic layer was washed with water and brine, dried over MgSO_4_, and concentrated. The residue was purified by flash column chromatography (CH_2_Cl_2_/MeOH 95/5) to give 0.11 g (27%) of the desired compound as a pink solid. Rf = 0.33 (CH_2_Cl_2_/MeOH 95/5); ^1^H NMR (400 MHz, MeOD) δ 8.88 (d, *J* = 2.0 Hz, 1H), 8.81–8.69 (m, 2H), 8.56 (d, *J* = 2.9 Hz, 1H), 8.22 (dd, *J* = 8.1, 2.1 Hz, 1H), 7.71 (dd, *J* = 8.9, 2.9 Hz, 1H), 4.45–4.25 (m, 2H), 4.10 (s, 2H), 4.04–3.93 (m, 2H), 3.50 (s, 4H), 3.33 (t, *J* = 1.6 Hz, 1H), 1.50 (s, 18H); ^13^C NMR (101 MHz, MeOD) δ 170.55, 163.15, 163.04, 158.24, 150.67, 148.89, 141.66, 139.29, 138.58, 138.49, 125.57, 123.75, 121.49, 81.08, 70.31, 59.99, 55.12, 54.74, 27.04.

Di-*tert*-butyl 2,2′-(((6-(6-(5-(2-(((4-nitrophenyl)sulfonyl)oxy)ethoxy)pyridin-2-yl)-1,2,4,5-tetrazine-3-yl)pyridin-3-yl)methyl)azanediyl)diacetate (**45a**)

The compound was obtained from **46** (0.05 g, 0.09 mmol) as reported for compound **3a**. Purification by flash chromatography (heptane/EtOAc 40/60) afforded 0.045 g (67%) of the desired compound as a red solid. Rf = 0.26 (heptane/EtOAc 30/70); ^1^H NMR (600 MHz, CDCl_3_) δ 8.82 (d, *J* = 2.1 Hz, 1H), 8.70–8.59 (m, 2H), 8.48 (d, *J* = 2.9 Hz, 1H), 8.35 (d, *J* = 8.8 Hz, 2H), 8.15–7.98 (m, 3H), 7.32 (d, *J* = 2.9 Hz, 1H), 4.76–4.47 (m, 2H), 4.41–4.31 (m, 2H), 4.01 (s, 2H), 3.40 (s, 4H), 1.41 (s, 18H); ^13^C NMR (151 MHz, CDCl_3_) δ 169.22, 162.64, 162.15, 155.32, 150.32, 149.95, 148.26, 142.30, 140.49, 138.34, 137.23, 137.03, 128.33, 124.54, 123.55, 123.24, 120.47, 80.37, 67.51, 65.00, 54.31, 53.61, 27.18; HPLC-MS [M+H]+ *m/z* calc. for [C_33_H_39_N_8_O_10_S]+: 739.25; found: 739.26.

### 3.2. Radiochemistry

#### 3.2.1. [^18^F]Fluoride Production and General Methods

The [^18^F]Fluoride was produced by a cyclotron CTI Siemens Eclipse, Rigshospitalet, Denmark, by irradiating [^18^O]H_2_O via a (p,n) reaction. Automated synthesis was performed on a Scanys synthesis module (Scansys Laboratorieteknik, Denmark), and analytical HPLC was performed on a Thermo Fisher UltiMate 3000 equipped with a C18 column (Luna 5 μm C18(2) 100 Å, 150 mm × 4.6 mm). Eluents: A, H_2_O with 0.1% TFA; B, MeCN with 0.1% TFA. Gradient: from 100% A to 100% B over 15 min, back to 100% A over 4 min, flow rate 1.5 mL/min. Detection by UV absorption was at λ = 254 nm on a UVD 170U detector, and radioactivity was analyzed with a flow-through GM-tube-based radiodetector (Scansys).

#### 3.2.2. Radiolabeling

The aqueous [^18^F]fluoride solution received from the cyclotron was passed through a preconditioned anion exchange resin (Sep-Pak Light QMA cartridge). The QMA was preconditioned by flushing it with 10 mL 0.5 M K_3_PO_4_ and washing it with 10 mL H_2_O afterward. The [^18^F]F^-^ was eluted from the QMA into a 4 mL v-shaped vial using 1 mL Bu_4_NOMs (20 µmol, 6,8 mg) dissolved in MeOH. The eluate was dried at 100 °C for 5 min under N_2_ flow. Compound **45a** was dissolved in 167 µL DMSO and then diluted with 833 µL *t*BuOH (1:5 ratio). The solution was added to the dried fluoride solution an allowed to react for 5 min at 100 °C. The reaction was cooled to 50 °C with air before addition of 3 mL H_2_O. This mixture was applied to a Sep-Pak C18 Plus solid phase extraction (SPE) cartridge that was preconditioned by flushing it with 10 mL EtOH followed by 10 mL of H_2_O. The SPE was flushed with another 5 mL of H_2_O and dried with N_2_. The product was eluted from the SPE with 2 mL MeCN into a 7 mL v-shaped vial containing 600 µL TFA. This mixture was reacted for 10 min at 80 °C. The reaction was then concentrated under N_2_ flow for 20 min to reduce the solvent volume to <0.1 mL. To this crude product mixture, 2.5 mL of H_2_O was added, and this solution was purified by semipreparative HPLC (Luna 5 μm C18(2) 100 Å, 250 mm × 10 mm, isocratic, 70% EtOH in H_2_O with 0.1% TFA 3 mL/min (rt: 13 min)). The product was collected in a 20 mL vial and diluted with 100 mM sterile phosphate buffer to adjust the pH to 5–8. The max EtOH concentration was 5%, and the activity concentration was 30–80 MBq/mL.

#### 3.2.3. Tetrazine Core Reactivity Test

The reaction between [^18^F]**45** and TCO-PNB was performed by mixing the formulated [^18^F]**45** (200 µL) with 5 µL of the commercially available TCO-PNB ester dissolved in DMF (5 mg/mL) in an analytical HPLC vial. The solution was gently shaken and left for 1 min before it was injected into the analytical HPLC instrument for analysis.

### 3.3. Blocking Assay and Ex Vivo Studies

#### 3.3.1. Establishing Tumor Xenografts in Mice

All animal studies were approved by the Danish Animal Experiments Inspectorate, Ministry of Food, Agriculture and Fisheries of Denmark (license no. 2016-15-0201-00920).

The human colon cancer cell line LS174T (ATCC, Manassas, VA, USA) was cultured in minimum essential medium (MEM) supplemented with 10% fetal bovine serum, 1% *L*-glutamine, 1% sodium pyruvate, 1% nonessential amino acids, and 1% penicillin–streptomycin (all from Thermo Fisher Scientific, Waltham, MA, USA). The cells were trypsinized and harvested for inoculation when they were in their exponential growth phase.

Subcutaneous tumors were established in the flank of five-week-old female nude BALB/c mice (Janvier Labs, Le Genest-Saint-Isle, France) by inoculation of ~5 × 10^6^ LS174T cells (in 100 μL sterile PBS).

The tumor volume was estimated from caliper measurements using the formula: volume = ½ (length × width^2^).

#### 3.3.2. Blocking Experiments

The blocking experiments were performed as previously described [17,32]. Briefly, tumor-bearing animals were grouped based on their tumor volume (~100–300 mm^3^, *n* = 3/group) and administered 100 µg/100 µL of CC49-TCO per mouse (~7 TCO/mAb). The ability of nonradioactive Tzs to block the binding between [^111^In]**47** and CC49-TCO was evaluated three days later. First, the animals were injected with the (nonradioactive) Tzs (39 nmol) that were chosen for in vivo evaluation. After a lag time of 1 h, [^111^In]**47** (~5 MBq, 3.9 nmol) was administered, and after 22 h, the mice were euthanized. Tissues were resected and weighted, and the radioactivity was measured using a gamma counter (Wizard2, Perkin Elmer). Data were corrected for decay, tissue weight, and injected amount of radioactivity. The setup also included a control group of mice receiving the precursor of [^111^In]**47** instead of a test compound 1 h prior to [^111^In]**47** (positive control), as well as a group exclusively receiving [^111^In]**47**. The tumor uptake of the different evaluated Tzs was normalized to the tumor uptake of the latter to determine the blocking effect.

#### 3.3.3. Pretargeted Imaging

LS174T xenografts were established in mice as previously described. When tumors reached a size of ~100 mm^3^, animals were divided into two groups based on their tumor volume (*n* = 5/group), and injected iv with 50 μg in 100 μL of either CC49-TCO or CC49. After 72 h the animals were administered intravenously with [^18^F]**45** (4.74 ± 1.39 MBq in 100 μL of PBS) and scanned using PET/CT (Inveon, Siemens Medical Solutions, USA) 1 h later, using a PET acquisition time of 5 min, an energy window of 350–650 KeV, and a time resolution of 6 ns; followed by a continuous 360 projection/360° CT scan, acquired with an X-ray tube voltage of 65 kV, a tube anode current of 500 μA, and an exposure time per projection of 400 ms. Mice were anesthetized by breathing 3% sevoflurane (5% for induction) mixed in 35% O_2_ in N_2_, and during scans the body temperature of the mice was kept stable using a heating pad.

Signograms from PET scans were reconstructed using a three-dimensional maximum a posteriori algorithm with correction for scatter and attenuation. The mean percentage of injected dose per grams (%ID/mL) was determined by manually creating regions of interest (ROI) on coregistered PET/CT images (Inveon Research Workplace software, Siemens Medical Solutions, Malvern, PA, USA).

GraphPad Prism 9 (GraphPad Software, La Jolla, CA, USA) was used for statistical analysis, and an unpaired *t*-test with Welch’s correction was used to compare the tumor uptake in the two groups. Results were considered significant when *p* < 0.05.

## 4. Conclusions

In this study, we reported the development of the first bispyridyl Tz directly labeled with fluorine-18. The [^18^F]**45** was obtained with sufficient yield, purity, and molar activity for in vivo evaluation. This imaging agent had comparable performances and target-to-background ratios compared to previously reported successful Tz tracers. Further evaluation studies are needed in order to optimize its potential for pretargeted imaging. The developed [^18^F]**45** is of special interest in pretargeted imaging, as it may be used—as a bispyridyl-based pretargeted imaging agent—to quantify drug release based on ‘click-to-release’ approaches. Bispyridyls have recently been shown to near-quantitatively release drugs from iTCOs within <10 min [41]. This is so far the fastest release demonstrated. The ability to quantify in vivo drug release is essential to precisely fine-tune dosing—one of the most important factors influencing the effectiveness of a treatment.

## Data Availability

Data is contained within the article and Appendix A.

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
