# Peer review of "Development of 18F-Labeled Bispyridyl Tetrazines for In Vivo Pretargeted PET Imaging"

_pharmaceuticals, 2022, doi:10.3390/ph15020245_

Round 1
Reviewer 1 Report
The authors present their work on the development of 18F-labeled bispyridyl-tetrazines for pretargeted PET imaging. The work is an interesting extension of their prior work. The fluorine-18 labelling of tetrazine containing molcules has been a difficult challenge in the field and limited the use of fluorine-18 with this approach, so the results reported are of interest to the community and to the larger community of chemist in general given the challenge of working with tetrazines and fluoride. The authors study is well designed and described. I recommend this manuscript for publication after minor revisions.
Abstract: "These structures allow for an easier structural modification as reported directly 18F-labeled tetrazines and are as such a viable method to develop advanced bi-functional pretargeting tools." This sentence is awkward, please improve it to aide the readers understanding.
Page 1: “rather days than” should be “days rather than”
Page 3: How stable are the fluoroethyl prosthetic groups utilized? In the literature many of these prove to be unstable due to metabolism and as a result deuterium containing analogues have been developed (e.g. fluoroethyl tyrosine and others Kuchar, et. Al Methods to Increase the Metabolic Stability of 18F-Radiotracers Molecules. 2015 Sep; 20(9): 16186–16220. doi: 10.3390/molecules200916186 ).
In other work O-CH2CF3 has been utilized instead to avoid this issue (Kaur, et. Al. Synthesis and Evaluation of a Fluorine-18 Radioligand for Imaging Huntingtin Aggregates by Positron Emission Tomographic Imaging Front Neurosci. 2021; 15: 766176. doi: 10.3389/fnins.2021.766176); where those approaches considered?
Looking at table 1, stability of the fluoroethyl could be an issue to address by one of the previously mentioned methods, if you have interest in those other analogues.
Page 4: I am unclear what you mean by “…amines over ethers to amides” did you prefer esters to amides and esters over ethers or did you mean something else? Please make this more clear.
Figure 3: Why was that pyridine (ortho to tetrazine linkage) introduced as opposed to the pyridine nitrogen at other positions?
Table 1: Please add R to the table as a column, so it is easier to determine which analogue is which in the data.
Scheme 5: In the caption please list 39 where appropriate, so it is clear how that portion of the molecule is added in the scheme.
Methods, Compound 3: Did you mean to say, washed with a saturated aqueous solution of NH4Cl?
Methods: For the final products of interest, especially 45, was HR-MS obtained to further confirm identity?
Author Response
Comment: The authors present their work on the development of 18F-labeled bispyridyl-tetrazines for pretargeted PET imaging. The work is an interesting extension of their prior work. The fluorine-18 labelling of tetrazine containing molcules has been a difficult challenge in the field and limited the use of fluorine-18 with this approach, so the results reported are of interest to the community and to the larger community of chemist in general given the challenge of working with tetrazines and fluoride. The authors study is well designed and described. I recommend this manuscript for publication after minor revisions.
Answer: We thank Reviewer 1 for the nice comments.
Comment: Abstract: "These structures allow for an easier structural modification as reported directly 18F-labeled tetrazines and are as such a viable method to develop advanced bi-functional pretargeting tools." This sentence is awkward, please improve it to aide the readers understanding.
Answer: We agree with Reviewer 1. The sentence has been changed to “These structures allow an easy chemical modification of 18F-labeled tetrazines paving the road toward highly functionalized pretargeting tools”.
Comment: Page 1: “rather days than” should be “days rather than”
Answer: The sentence has been changed as suggested.
Comment: Page 3: How stable are the fluoroethyl prosthetic groups utilized? In the literature many of these prove to be unstable due to metabolism and as a result deuterium containing analogues have been developed (e.g. fluoroethyl tyrosine and others Kuchar, et. Al Methods to Increase the Metabolic Stability of 18F-Radiotracers Molecules. 2015 Sep; 20(9): 16186–16220. doi: 10.3390/molecules200916186 ). In other work O-CH2CF3 has been utilized instead to avoid this issue (Kaur, et. Al. Synthesis and Evaluation of a Fluorine-18 Radioligand for Imaging Huntingtin Aggregates by Positron Emission Tomographic Imaging Front Neurosci. 2021; 15: 766176. doi: 10.3389/fnins.2021.766176); where those approaches considered? Looking at table 1, stability of the fluoroethyl could be an issue to address by one of the previously mentioned methods, if you have interest in those other analogues.
Answer: No, we never considered this approach, but we agree with the reviewer. In some cases, the fluoroethyl group is not metabolically stable. However due to the unique properties of the pretargeting approach we think that this is not an issue. As a matter of fact, tetrazines should reach their target and “click” with the TCO-modified antibodies within minutes. This should prevent the metabolism of the “clicked” tetrazines. On the other hand, the unreacted tetrazine in the bloodstream is designed to be extremely polar and is excreted very quickly. Moreover, the tetrazine core will most likely still remain the easier moiety to be metabolized. For all these reasons we believe that in this specific case metabolism of the fluoroethyl group should not be an issue.
Comment: Page 4: I am unclear what you mean by “…amines over ethers to amides” did you prefer esters to amides and esters over ethers or did you mean something else? Please make this more clear.
Answer: The sentence has been changed to “such as esters, amines, ethers and amides”.
Comment: Figure 3: Why was that pyridine (ortho to tetrazine linkage) introduced as opposed to the pyridine nitrogen at other positions?
Answer: There is no real chemical reason. We usually draw bis pyridyl tetrazines in this way. These compounds are reported in this way in many other articles (e.g. Raffaella Rossin, Sander M. J. van Duijnhoven, Tilman Läppchen, Sandra M. van den Bosch, and Marc S. Robillard, Molecular Pharmaceutics 2014 11 (9), 3090-3096 DOI: 10.1021/mp500275a)
Comment: Table 1: Please add R to the table as a column, so it is easier to determine which analogue is which in the data.
Answer: R has been added to the Table as requested.
Comment: Scheme 5: In the caption please list 39 where appropriate, so it is clear how that portion of the molecule is added in the scheme.
Answer: We agree with the reviewer. However, 39 and 40 are listed already in the conditions ii) and iv) of the synthetic scheme.
Comment: Methods, Compound 3: Did you mean to say, washed with a saturated aqueous solution of NH4Cl?
Answer: We agree with the reviewer. The sentence has been changed as suggested.
Comment: Methods: For the final products of interest, especially 45, was HR-MS obtained to further confirm identity?
Answer: We did not perform HRMS analysis since it was not formally required from the journal author guidelines. Unfortunately, we don t have easy access to HRMS and it would take longer than 10 days required for the revision. However, we obtained the HPLC-MS data for the final compounds 41, 45 and for the key intermediate 45a (added in the text).
Reviewer 2 Report
The manuscript describes the investigation of suitable precursors to obtain 18F-labeled bispyridinyl tetrazines. After obtaining a suitable precursor, the authors turn it into a more hydrophilic derivative in order to putatively improve its pharmacokinetics, and briefly studied its applicability for pretargeted PET imaging.
The work is very timely, as pretargeting is getting more and more traction in nuclear medicine. The first part of the study, concerning the development of efficient precursors, is well designed and provides an interesting set of results for the future development of 18F-labeled tetrazines. However, the in vivo characterization of the developed tetrazines is very incomplete, and in several occurrences the results are not sufficient to support the claims of the authors.
The authors need to address the following points.
- On the writing level, improvements could be done. Especially, the purpose of a figure caption is to explain and somewhat describe the figure, not to discuss the how and the why of the project or experiment, as it happens for fig 1, fig 2, fig 3, Table 2, etc. In each of these captions, the reader will find either important concepts belonging to the main text, or information already mentioned in the main text and proving useless to the understanding of the figure/table. Similarly, experimental details such as HPLC conditions should be left to the Materials and Methods section.
- There are frequent confusions throughout the manuscript between the fluor-labeled tetrazines and their precursors (typical example page 8), making it sometimes difficult to follow.
- In figure 1, the authors state that conventional immunoimaging yields low contrast, while pretargeted immunoimaging yield high contrast. How so? For the vast majority of radiolabeled tetrazines, much higher contrasts are obtained with direct labelling of the antibodies (e.g. the most promising compound in the present study, with a tumor-to-heart ratio of 0.8). Please elaborate.
- In section 2, all the subsections are numbered 2.1.
- Table 1 must be moved to the corresponding section (2.4.) and compound 4 removed, as no labelling experiment was performed on this compound. Please add a column with the R group next to the X group. And please provide the concentration of precursors rather than the amount. Please remove footnotes e and f, and analytical conditions of the HPLC.
- In table 2, please provide standard deviations for the blocking effect and tumor uptake of 111In-47. Please remove n.d. in the notes. Define TPSA.
In section 2.6.
- Please describe the synthesis of the precursor 45a.
- Concerning the blocking experiment, I understand that the optimization of the radiolabeling of all developed tetrazine is tedious work. However, in this study, only one tetrazine was selected for in vivo study (tetrazine 45). Therefore, the use of this assay seems useless. Going directly to PET imaging and comparing with tetrazine 47 would have reduced the number of animals, and provided more interesting results. The assay developed by the author has also a huge drawback: it provides information only about the on-target coupling, eluding off-target coupling and accumulation in non-target tissues. Tumor uptake is just one aspect influencing the imaging sensitivity, much more important are the tumor-to-background ratios (see below).
- Was the blocking with tetrazine 47 performed with indium-chelating 47 or only 47? The resulting variation in global charge between indium-chelating 47 and free 47 will surely make a huge difference in terms of pharmacokinetics.
- The authors should perform a negative control by injecting only 111In-47 (without CC49-TCO) in order to eliminate non-specific uptake of the tracer in the tumor when calculating the blocking effect.
- In section 2.7, figure 4C does not demonstrate that 18F-45 is stable for more than 4 hours. Please provide the pH of the PBS used for this stability study.
Section 2.8.
- The authors must explain their choice of limiting the image-derived biodistribution study to only tumor, blood and muscle. The limiting factor of pretargeting tracers is more than often the non-specific accumulation of the tracer in viscera, so why not studying the activity uptake in these organs as well?
- Using the activity in the heart as a surrogate for activity in the blood can be a dangerous shortcut. The authors must provide references demonstrating that there is no uptake of CC49 in the heart.
- The authors very easily interchange tumor-to-blood and tumor-to-muscle with tumor-to-background, but this is true only for tumors located in the heart or in the muscle. For the vast majority of cancer, the background will be the viscera. The authors need to provide activity concentration in the main organs and calculate tumor-to-background ratios based on these values.
- Image-derived activity concentration must be expressed in %ID/mL or %ID/cm3.
- In figure 5, please specify if the images are slices or maximum intensity projections (MIP). Please provide MIPs in the SI. In figure 5C, please provide the T/B and T/M ratios for the negative control (CC49). Add error bars in figure 5C.
- In the Materials and methods section, the authors must provide information about the synthesis and characterization of the CC49-TCO conjugate.
- In section 3.3.3., the following sentence is unclear: “animals were matched based on their tumor volume and divided in two groups”. Could the authors explain?
- In this same section, specify the buffer used for the injection of the antibody.
- In the conclusion, the authors claim that 18F-45 has comparable pharmacokinetics than previously reported tracers. Yet, the pharmacokinetics of 18F-45 are not studied nor described in the present manuscript.
- Several abbreviations are used in the manuscript without being defined (eg. NIR, TPSA, TCO-PNP, EOS, TCO-PNB, Bu4NOMs). Please also add an abbreviation list at the end of the manuscript.
Here is also a non-exhaustive list of mistakes and typo:
In vitro and in vivo in italic.
Page 5: these intermediates were; were obtained in…
Page 6: With the exception of compound 3, were in the range of , were designed, synthesized
Page 7: analogues of 3; 6-Cyanonicotinic;
Page 8: were in the range of; an easy-to-label compound
In the supplementary :
- Please provide captions for all the figures.
- Compound 45a is not pure (impurities at 3.5, 7.5, 8.5, 9.5 min…). Integrate the UV trace correctly, or remove the integration and the 100% purity claim.
Author Response
Reviewer #2:’
Comment: The manuscript describes the investigation of suitable precursors to obtain 18F-labeled bispyridinyl tetrazines. After obtaining a suitable precursor, the authors turn it into a more hydrophilic derivative in order to putatively improve its pharmacokinetics, and briefly studied its applicability for pretargeted PET imaging. The work is very timely, as pretargeting is getting more and more traction in nuclear medicine. The first part of the study, concerning the development of efficient precursors, is well designed and provides an interesting set of results for the future development of 18F-labeled tetrazines. However, the in vivo characterization of the developed tetrazines is very incomplete, and in several occurrences the results are not sufficient to support the claims of the authors.
Answer: We thank reviewer 2 for the comments.
Comment: The authors need to address the following points. On the writing level, improvements could be done. Especially, the purpose of a figure caption is to explain and somewhat describe the figure, not to discuss the how and the why of the project or experiment, as it happens for fig 1, fig 2, fig 3, Table 2, etc. In each of these captions, the reader will find either important concepts belonging to the main text, or information already mentioned in the main text and proving useless to the understanding of the figure/table. Similarly, experimental details such as HPLC conditions should be left to the Materials and Methods section.
Answer: We agree with the reviewer, the captions in the figures have been reduced as requested. However, we think that for some figures (eg Fig 2) some additional explanations are needed.
Comment: There are frequent confusions throughout the manuscript between the fluor-labeled tetrazines and their precursors (typical example page 8), making it sometimes difficult to follow.
Answer: We agree with the reviewer. We modified paragraph 2.5 to make it clearer. In general, all the precursors are referred with the compound number followed by “a”. The radiolabeled compounds are named “[18F]” followed by the compound number. We believe it is easy to follow this numbering.
Comment: In figure 1, the authors state that conventional immunoimaging yields low contrast, while pretargeted immunoimaging yield high contrast. How so? For the vast majority of radiolabeled tetrazines, much higher contrasts are obtained with direct labelling of the antibodies (e.g. the most promising compound in the present study, with a tumor-to-heart ratio of 0.8). Please elaborate.
Answer: We agree with the reviewer. When we consider later timepoints the accumulation of the directly labeled antibody is much higher. However, in the figure we state that this is true at early timepoints because when the labelled mAb has distributed (for days) the imaging contrast is good and definitely not low. However, with regard to imaging, the pre-targeting approach makes it possible to scan early after administration of radioligand and allows for use of short-lived isotopes optimal for imaging like F-18 (thereby also lowering the radiation dose). Furthermore, we agree with the reviewer that the low tumor-to-blood ratio is a limitation. This phenomeon, most likely caused by binding of radiolabeled Tz to antibodies in the circulation, is a general challenge for the pretargeting approach. This issue could be solved employing masking agents or clearing agents. Anyway, in order to avoid confusion, the caption from Figure 2 has been removed (as asked above).
‘
Comment: In section 2, all the subsections are numbered 2.1.
Answer: We thank the reviewer. Most likely there was a problem during MDPI formatting.
Comment: Table 1 must be moved to the corresponding section (2.4.) and compound 4 removed, as no labelling experiment was performed on this compound. Please add a column with the R group next to the X group. And please provide the concentration of precursors rather than the amount. Please remove footnotes e and f, and analytical conditions of the HPLC.
Answer: We agree with Reviewer. We made all the changes requested. However, we think that the table should be stay in section 2.3 where the labeling is firstly described.
Comment: In table 2, please provide standard deviations for the blocking effect and tumor uptake of 111In-47. Please remove n.d. in the notes. Define TPSA.
Answer: We agree with the reviewer. The changes have been made as requested and standard deviations have been added.
Comment: In section 2.6. Please describe the synthesis of the precursor 45a.
Answer: We agree with the reviewer. The description has been added as suggested (section 2.7).
Comment: Concerning the blocking experiment, I understand that the optimization of the radiolabeling of all developed tetrazine is tedious work. However, in this study, only one tetrazine was selected for in vivo study (tetrazine 45). Therefore, the use of this assay seems useless. Going directly to PET imaging and comparing with tetrazine 47 would have reduced the number of animals, and provided more interesting results. The assay developed by the author has also a huge drawback: it provides information only about the on-target coupling, eluding off-target coupling and accumulation in non-target tissues. Tumor uptake is just one aspect influencing the imaging sensitivity, much more important are the tumor-to-background ratios (see below).
Answer: This method gives us a general indication about the in vivo performance of a tetrazine. This assay has been previously employed on a large set of tetrazines (almost 50 compounds) proving to be able to predict if a tetrazine will be a good pretargeting tool (Stéen, E.J.L.; Jørgensen, J.T.; Denk, C.; Battisti, U.M.; Nørregaard, K.; Edem, P.E.; Bratteby, K.; Shalgunov, V.; Wilkovitsch, M.; Svatunek, D.; et al. Lipophilicity and Click Reactivity Determine the Performance of Bioorthogonal Tetrazine Tools in Pretargeted In Vivo Chemistry. ACS Pharmacology & Translational Science 2021, 4, 824-833, doi:10.1021/acsptsci.1c00007). From this assay we discovered that the tumour accumulation and the tumor-to-background ratios are somehow related and connected with polarity. For this reason, we tested compound 41 (hydrophobic analogue) and 45 (hydrophilic analogue). The results obtained were completely in line with our previous findings. Furthermore, since tetrazines are quite labile compounds other factors need to be taken into account like stability, metabolism of the reactive tetrazine core and so on. The blocking assay is able to give us a general and broad view on the capability of the tetrazine to reach the tumor and still “click” with the TCOs. We believe that these data are maybe not necessary but still useful before performing a PET experiment.
Comment: Was the blocking with tetrazine 47 performed with indium-chelating 47 or only 47? The resulting variation in global charge between indium-chelating 47 and free 47 will surely make a huge difference in terms of pharmacokinetics. The authors should perform a negative control by injecting only 111In-47 (without CC49-TCO) in order to eliminate non-specific uptake of the tracer in the tumor when calculating the blocking effect.
Answer: We agree with the reviewer that chelated and non-chelated compounds might have different PK. The assay was done using [111In]47 since we were using radioactivity to evaluate the blocking. Compound 47 (non chelated) was added only as a control to verify that the tumor uptake of [111In]47 could indeed be blocked. Moreover, the blocking effect is not an absolute but relative measure able to compare tumor accumulation of tetrazines. Like when performing preclinical imaging studies using tumor mouse models, the assumption is that groups of mice, for example bearing LS174T xenograft tumor of similar size grown subcutaneously, will have approximately the same tumor uptake, as well as non-specific accumulation of the test compounds. If this was not the case it would not make sense to compare groups at all, but of course the group size of animals needed to obtain this presumption can always be discussed.
Comment: In section 2.7, figure 4C does not demonstrate that 18F-45 is stable for more than 4 hours. Please provide the pH of the PBS used for this stability study.
Answer: Figure 4C shows that the tetrazine core is stable for 4 hours since it still able to react with a TCO-PNB forming the corresponding dihydropyridazines (multiple tautomers/positional isoemers). This is a standard way to demonstrate that the tetrazine reactivity is present and that all the radiolabeled compound can still be clicked with the corresponding IEDDA partner. The pH has been added as requested.
Comment: Section 2.8. The authors must explain their choice of limiting the image-derived biodistribution study to only tumor, blood and muscle. The limiting factor of pretargeting tracers is more than often the non-specific accumulation of the tracer in viscera, so why not studying the activity uptake in these organs as well?
Answer: We agree with the reviewer. However, the most frequently used tumor-to-background ratios for PET imaging is tumor-to-blood and tumor-to-muscle. The former because it gives a measure between the tumor and circulating tracer; the later one because it is representative for the image contrast to the organ system with the largest volume. Tumor-to-liver contrast could be added since it represents a tissue with high accumulation. In previous experiments we observe for these very polar tetrazines a quick excretion so we believe that this value would not be relevant for the manuscript. However, if the reviewer considers the data necessary, we are keen to add them.
Comment: Using the activity in the heart as a surrogate for activity in the blood can be a dangerous shortcut. The authors must provide references demonstrating that there is no uptake of CC49 in the heart.
Answer: We agree that the heart cannot be used as a surrogate for blood uptake in cases where the target is found in the heart. However, this is not the case for TAG-72. Ex vivo biodistribution of CC49-TCO + [111In]47 can be found in our recent published paper (García-Vázquez, R.; Battisti, U.M.; Jørgensen, J.T.; Shalgunov, V.; Hvass, L.; Stares, D.L.; Petersen, I.N.; Crestey, F.; Löffler, A.; Svatunek, D.; et al. Direct Cu-mediated aromatic 18F-labeling of highly reactive tetrazines for pretargeted bioorthogonal PET imaging. Chemical Science 2021, doi:10.1039/D1SC02789A).
Comment: The authors very easily interchange tumor-to-blood and tumor-to-muscle with tumor-to-background, but this is true only for tumors located in the heart or in the muscle. For the vast majority of cancer, the background will be the viscera. The authors need to provide activity concentration in the main organs and calculate tumor-to-background ratios based on these values.
Answer: As for the comment above, the standard way to present PET data is T/B and T/M ratio. However, we could add T/L ratio if the reviewer thinks it is necessary.
Comment: Image-derived activity concentration must be expressed in %ID/mL or %ID/cm3.
Answer: We agree with the reviewer. %ID/g has been changed to %ID/mL in all the text and figures as requested.
Comment: In figure 5, please specify if the images are slices or maximum intensity projections (MIP). Please provide MIPs in the SI. In figure 5C, please provide the T/B and T/M ratios for the negative control (CC49). Add error bars in figure 5C.
Answer: We think the T/B and T/M ratio for the negative control is not meaningful as the purpose is to show a difference between tumor uptake and a reference tissue. However, there is no tumor uptake in the animals pretreated with CC49, wherefore the ratios are not meaningful.
Comment: In the Materials and methods section, the authors must provide information about the synthesis and characterization of the CC49-TCO conjugate.
Answer: The CC49-TCO conjugate was provided by Tagworks Pharmaceuticals. A sentence and a reference have been provided in the general part of the experimental section
Comment: In section 3.3.3., the following sentence is unclear: “animals were matched based on their tumor volume and divided in two groups”. Could the authors explain? In this same section, specify the buffer used for the injection of the antibody.
Answer: We agree the sentence has been changed. The buffer has been added.
Comment: In the conclusion, the authors claim that 18F-45 has comparable pharmacokinetics than previously reported tracers. Yet, the pharmacokinetics of 18F-45 are not studied nor described in the present manuscript.
Answer: We agree with the reviewer. The term pharmacokinetic has been changed to performance.
Comment: Several abbreviations are used in the manuscript without being defined (eg. NIR, TPSA, TCO-PNP, EOS, TCO-PNB, Bu4NOMs). Please also add an abbreviation list at the end of the manuscript.
Answer: The abbreviations have been added as requested.
Comment: Here is also a non-exhaustive list of mistakes and typo: In vitro and in vivo in italic; Page 5: these intermediates were; were obtained in…; Page 6: With the exception of compound 3, were in the range of , were designed, synthesized; Page 7: analogues of 3; 6-Cyanonicotinic;; Page 8: were in the range of; an easy-to-label compound
Answer: We thank the reviewer. The typos and the error have been corrected. The manuscript was checked for additional mistakes.
Comment: In the supplementary : Please provide captions for all the figures.
Answer: The captions have been added for all the figures.
Comment: Compound 45a is not pure (impurities at 3.5, 7.5, 8.5, 9.5 min…). Integrate the UV trace correctly, or remove the integration and the 100% purity claim.
Answer: The UV-trace has been correctly integrated as suggested.
Round 2
Reviewer 2 Report
The authors have done a good job editing and correcting the manuscript. However, there are still a few points where the answers and justifications provided by the authors are not convincing.
Figure 1: with this 72h delay on conventional immunoimaging, one would achieve good contrast. Please remove this 'low contrast' statement, as it is very deceiving. Also, both conventional and preT are tagged as A)
Figure 4C caption must be edited to mention this 4h incubation.
Fig 5: Please add the tumor-to-liver ratio, at the very least.
Fig 5: The point of the authors concerning the ratios of the negative control is not valid. There is a significant tumor uptake in the negative control (0.4 %ID/mL, almost a forth of the tumor uptake obtained with pretargeting). The authors must provide tumor-to-background (blood, muscle and liver) for the negative control as well (CC49).
Author Response
Comment: The authors have done a good job editing and correcting the manuscript. However, there are still a few points where the answers and justifications provided by the authors are not convincing.
Answer: We thank the reviewer for the comment.
Comment: Figure 1: with this 72h delay on conventional immunoimaging, one would achieve good contrast. Please remove this 'low contrast' statement, as it is very deceiving. Also, both conventional and preT are tagged as A)
Answer: We agree with the reviewer. The figure has been changed as requested. Now for conventional imaging it is written “high contrast after days” while in pretargeting imaging it is written “high contrast after hours”.
Comment: Figure 4C caption must be edited to mention this 4h incubation.
Answer. The caption has been changed accordingly.
Comment: Fig 5: Please add the tumor-to-liver ratio, at the very least.
Answer: The liver uptake in the groups has been added to fig 5. In order to fit the liver uptake in the figure the ratios have been removed from the figure and are now in the text.
Comment: Fig 5: The point of the authors concerning the ratios of the negative control is not valid. There is a significant tumor uptake in the negative control (0.4 %ID/mL, almost a forth of the tumor uptake obtained with pretargeting). The authors must provide tumor-to-background (blood, muscle and liver) for the negative control as well (CC49).
Answer: We thank the reviewer for the comment. The T/M-ratio of the control group is 5.4, the T/B-ratio is 0.5 and the T/L-ratio is 0.05. However, we still don’t think it is meaningful to compare the ratios between pretargeted animals and controls in this study. The situation here is different from conventional imaging that normally includes a tissue of interest with high expression of a target and some reference tissue. A control group will typically then have reduced expression of the target or some kind of blocking will be performed. However, in this situation the control represents the biodistribution of 18F-labbelled Tz when no TCOs is present and the bio-distribution of the animals pretreated with CC49-TCO, represents a “mixture” between “free” 18F-labeled Tz and 18F-labeled Tz conjugated to the antibody. Also, as there is still a lot of circulating antibody at the time of Tz administration, both uptake in target and reference tissue will change, which also further complicates the interpretation when comparing the ratios. However, we will of course be willing to report the numbers in case it is requested.